# A neural circuit for flexible control of persistent behavioral states

**Ni Ji[1], Gurrein K Madan[1], Guadalupe I Fabre[1], Alyssa Dayan[1], Casey M Baker[1], Talya S Kramer[1,2], Ijeoma Nwabudike[1], Steven W Flavell[1]\***

[1]Picower Institute for Learning & Memory, Department of Brain & Cognitive Sciences, Massachusetts Institute of Technology, Cambridge, United States; [2]MIT Biology Graduate Program, Massachusetts Institute of Technology, Cambridge, United States

**Abstract** To adapt to their environments, animals must generate behaviors that are closely aligned to a rapidly changing sensory world. However, behavioral states such as foraging or courtship typically persist over long time scales to ensure proper execution. It remains unclear how neural circuits generate persistent behavioral states while maintaining the flexibility to select among alternative states when the sensory context changes. Here, we elucidate the functional architecture of a neural circuit controlling the choice between roaming and dwelling states, which underlie exploration and exploitation during foraging in *C. elegans*. By imaging ensemble-level neural activity in freely moving animals, we identify stereotyped changes in circuit activity corresponding to each behavioral state. Combining circuit-wide imaging with genetic analysis, we find that mutual inhibition between two antagonistic neuromodulatory systems underlies the persistence and mutual exclusivity of the neural activity patterns observed in each state. Through machine learning analysis and circuit perturbations, we identify a sensory processing neuron that can transmit information about food odors to both the roaming and dwelling circuits and bias the animal towards different states in different sensory contexts, giving rise to context-appropriate state transitions. Our findings reveal a potentially general circuit architecture that enables flexible, sensory-driven control of persistent behavioral states.

**\*For correspondence:**
flavell@mit.edu

**Competing interest:** The authors declare that no competing interests exist.

## Editor's evaluation

In this study Ji and colleagues investigate the circuit mechanisms that control long lasting behavioral foraging states in the nematode *C. elegans*. In a series of elegant neuronal imaging, circuit manipulation and behavioral genetics experiments they show how populations of neurons tightly coordinate their ensemble activity to achieve a surprising degree of multi-functionality. This includes the control of locomotion parameters that characterize each foraging state, their mutual exclusiveness and persistence, as well as sensory integration for informed state switching.

## Introduction

The behavioral state of an animal—whether it is active, inactive, mating, or sleeping—influences its perception of and response to the environment (*Anderson, 2016*; *Artiushin and Sehgal, 2017*; *Lee and Dan, 2012*; *Maimon, 2011*; *Nichols et al., 2017*). In contrast to fast motor actions, behavioral states are often highly stable, lasting from minutes to hours. Despite this remarkable stability, animals can flexibly choose their behavioral state based on the sensory context and switch states when the context changes (*Andalman et al., 2019*; *Clowney et al., 2015*; *Hoopfer et al., 2015*). How the brain generates persistent behavioral states while maintaining the flexibility to select among alternative states is not well understood.

At the neural level, persistent behavioral states are often associated with stable patterns of neural activity. For example, continuous activation of pCd neurons in male *Drosophila* underlies persistent courtship and aggressive behaviors (*Jung et al., 2020*). In addition, recent large-scale recordings of neural activity have revealed that behavioral states such as sleep and active locomotion are represented as stable, stereotyped activity patterns in neurons spanning multiple brain regions (*Nichols et al., 2017*; *Andalman et al., 2019*; *Dunn et al., 2016*; *Kato et al., 2015*; *Marques et al., 2020*; *Musall et al., 2019*). While the encoding of a behavioral state can be broadly distributed, the neurons that control the onset and duration of a state are often a smaller subset of those that comprise the full circuit (*Andalman et al., 2019*; *Cho and Sternberg, 2014*). To gain mechanistic insights into how persistent behavioral states are generated and controlled, it will be critical to elucidate the functional interactions among key control neurons and understand how they incorporate incoming sensory inputs that influence behavioral states.

Past studies have proposed recurrent circuitry and neuromodulation as two central mechanisms that contribute to the generation of persistent behavioral states. While theoretical studies have shown that recurrent excitatory or inhibitory feedback can underlie stable firing patterns (*Machens et al., 2005*; *Major and Tank, 2004*; *Seung, 1996*; *Wang, 2001*), direct experimental evidence linking recurrent circuitry with persistent activity during minutes-long behavioral states remains scarce (the role of recurrent inhibition in fast timescale switching is better established *Wang, 2020*; *Roberts et al., 2016*). Neuromodulators are known to control persistent behaviors like sleep and wake states, as well as states of stress and hunger (*Bargmann, 2012*; *Marder, 2012*; *Saper et al., 2010*; *Seo et al., 2019*). However, our understanding of how ongoing neuromodulator release in vivo promotes persistent circuit activity remains limited. In addition, it is unclear how dynamic sensory inputs interact with recurrent circuitry and neuromodulation to elicit behavioral state transitions in a changing sensory environment.

In this study, we investigate the neural circuit mechanisms that give rise to circuit-level activity patterns during persistent foraging states in *C. elegans*. While foraging on bacterial food, *C. elegans* alternate between roaming states, characterized by high-speed forward movement and occasional reorientations, and dwelling states, marked by slow forward and backward movements (*Ben Arous et al., 2009*; *Fujiwara et al., 2002*; *Flavell et al., 2020*; *Kim and Flavell, 2020*). Each state can last up to tens of minutes and the transitions between states are abrupt. The fraction of time an animal spends in each state is influenced by its satiety, ingestion of bacterial food, and sensory cues such as odors (*Ben Arous et al., 2009*; *Fujiwara et al., 2002*; *Chew et al., 2018*; *Shtonda and Avery, 2006*; *Rhoades et al., 2019*). Consistent with the notion that these states reflect an exploration-exploitation tradeoff, animals favor dwelling in food-rich environments and after starvation, but favor roaming in poor-quality food environments and after aversive stimulation.

We and others previously found that serotonin (5-HT) and the neuropeptide pigment-dispersing factor (PDF) act as opposing neuromodulators that stabilize dwelling and roaming states, respectively (*Rhoades et al., 2019*; *Choi et al., 2013*; *Flavell et al., 2013*; *Sawin et al., 2000*; *Iwanir et al., 2016*). Serotonin acts through the serotonin-gated chloride channel MOD-1 to promote dwelling, with a smaller contribution from the other serotonin receptors (*Flavell et al., 2013*). PDF-1 and –2 neuropeptides act through a single PDF receptor, PDFR-1, to drive roaming (*Flavell et al., 2013*). Cell-specific genetic perturbations uncovered the neurons that produce and detect these neuromodulators to control the stability of each behavioral state (*Flavell et al., 2013*). However, these identified neurons are densely interconnected with one another and with other neurons in the *C. elegans* connectome, making it infeasible to infer the core functional circuitry that shapes the roaming and dwelling states from these previous genetic studies. Crucially, it remains unclear how 5-HT and PDF impact overall circuit activity to promote persistent behavioral states. In addition, while it is known that the sensory environment can influence roaming and dwelling, how sensory inputs converge onto this core neuromodulatory circuit to influence behavioral states remains an open question.

To address these questions, we performed simultaneous calcium imaging of defined neurons throughout the roaming-dwelling circuit in freely moving animals. We identified stereotyped, circuit activity patterns corresponding to each foraging state. By combining circuit imaging with genetic perturbations, we identified a mutual inhibitory loop between the serotonergic NSM neuron and the 5-HT and PDF target neurons. We found that this mutual inhibition is critical for the persistence and mutual exclusivity of the neural activity patterns observed during roaming and dwelling. Furthermore,

we found that the AIA sensory processing neuron sends parallel outputs to both neuromodulatory systems and can bias the circuit towards either roaming or dwelling, depending on the overall sensory context. Together, these results identify a functional circuit architecture that allows for flexible, sensory-driven control of persistent behavioral states.

## Results

### Roaming and dwelling states are associated with stereotyped changes in circuit activity

To understand how roaming and dwelling states arise from circuit-level interactions between neurons, we sought to monitor the activity of neurons throughout the core roaming-dwelling circuit in wild-type animals and additionally during perturbations that alter signaling among the neurons. We built a calcium imaging platform with a closed-loop tracking system that allows for simultaneous imaging of many neurons as animals freely move (*Figure 1A* and *Figure 1—figure supplement 1A-B*; *Faumont and Lockery, 2006*; *Nguyen et al., 2016*; *Venkatachalam et al., 2016*). We generated a transgenic line where well-defined promoter fragments were used to express GCaMP6m in a select group of 10 neurons (*Figure 1B*; *Figure 1—figure supplement 1C-D*; *Figure 1—figure supplement 2*). These neurons were selected based on their classification into at least one of the three following groups: (1) neurons expressing 5-HT, PDF, or their target receptors MOD-1 or PDFR-1 (*Flavell et al., 2013*), (2) neurons that share dense synaptic connections with those in group 1, and (3) premotor or motor neurons whose activities are associated with locomotion (*Roberts et al., 2016*; *Li et al., 2014*). A small subset of neurons that had been implicated in roaming and dwelling states were omitted from the multi-neuron GCaMP6m line because their cell bodies were not located in the head (HSN, PVP *Flavell et al., 2013*). We performed circuit-level imaging (at a volume rate of 2 Hz) of these transgenic animals as they foraged on uniformly seeded bacterial lawns (*Figure 1C* and *Figure 1—figure supplement 3*). Imaging this defined subset of neurons allowed us to leverage prior knowledge and easily determine the identity of each neuron in each recording, thereby circumventing the challenge of determining neuronal identity in a densely labeled brain.

While dwelling, animals move forwards at low speed and frequently display short, low-speed reversals. Roaming animals travel at high speed in forward runs that are punctuated by long, high-speed reversals, which change the animal's heading direction. To determine how neural activity in the roaming-dwelling circuit encodes locomotion parameters and/or behavioral states, we first examined whether each neuron's activity was associated with the animal's axial velocity (i.e. velocity projected along the body axis), axial speed, movement direction (forward or reverse) or foraging state (roaming or dwelling; a Hidden Markov Model used to segment roaming and dwelling states is described in the Methods; see also *Figure 1—figure supplement 4*). Six of the 10 recorded neurons displayed calcium levels that correlated with axial velocity and movement direction (*Figure 1D–E*; *Figure 1—figure supplement 5A*). Consistent with previous reports, we observed that the PDF-1-expressing neuron AVB and PDFR-1-expressing neurons AIY and RIB, known to promote forward runs (*Kato et al., 2015*; *Roberts et al., 2016*; *Li et al., 2014*; *Tsalik and Hobert, 2003*; *Gray et al., 2005*; *Luo et al., 2014*), exhibited increased activity during forward runs, while the premotor neuron AVA, known to promote reversals (*Kato et al., 2015*; *Roberts et al., 2016*; *Luo et al., 2014*; *Chalfie et al., 1985*), exhibited heightened activity during reversals. A partially overlapping group of neurons, including the serotonergic neuron NSM, displayed activity that co-varied with animal speed (*Figure 1—figure supplement 5A*). Moreover, we found that nine of the ten recorded neurons exhibited changes in activity as animals transitioned between roaming and dwelling states (*Figure 1D–E*). Interestingly, almost all neurons whose activity correlated with axial velocity and/or movement direction, in particular AVB, AIY, RIB, and AVA, exhibited reduced activity during dwelling, compared to roaming (overall and/or surrounding moments of state transitions; *Figure 1D–E*). This observation is consistent with the known roles of these neurons in driving locomotion, insofar as locomotion is reduced during dwelling. These effects can also be clearly detected when comparing the joint activity distributions of forward- and reverse-active neurons during roaming versus dwelling (*Figure 1—figure supplement 6*). In contrast, the serotonergic neuron NSM was more active during dwelling. Together, these data reveal that changes in the roaming/dwelling state of the animal are accompanied by changes in the

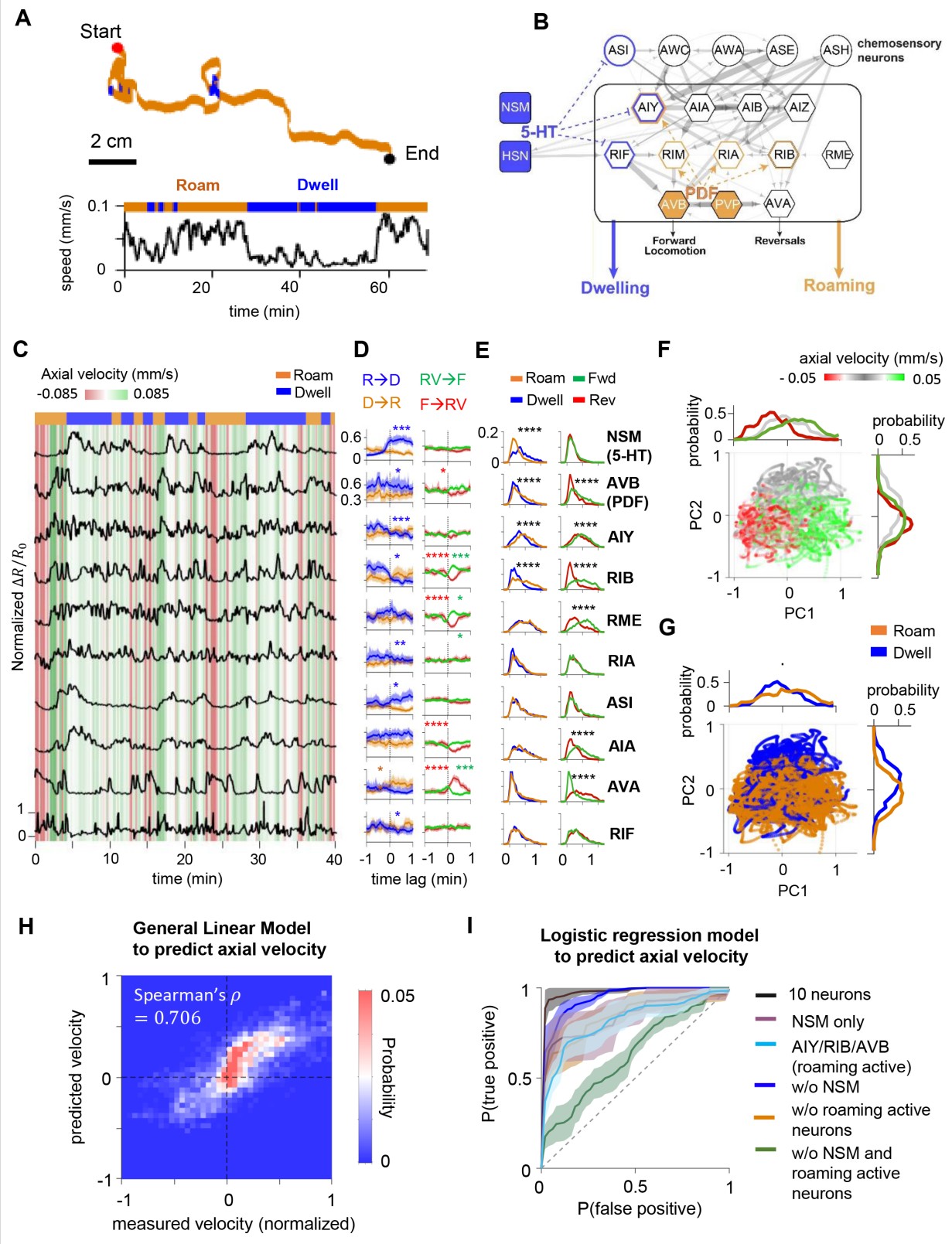

**Figure 1.** Circuit-wide calcium imaging reveals a stable, low-dimensional neural representation of foraging states. (**A**) (Top) Movement trajectory of a *C. elegans* animal foraging on bacterial food under the tracking microscope. Red and black dots mark the beginning and end of the trajectory, respectively. Orange indicates that the animal was in the roaming state, while blue indicates the animal was in dwelling state. (Bottom) The speed of the animal during the same period. (**B**) Putative neural circuit that mediates the sensory control of the roaming and dwelling states, based on the *C*.

*Figure 1 continued on next page*

*Figure 1 continued*

*elegans* connectome and genetic analyses from a previous study (***Flavell et al., 2013***). Each *C. elegans* neuron has a three-letter name. Blue highlights indicate sites of serotonin signaling and orange highlights indicate sites of PDF signaling. Gray arrows are synapses from the *C. elegans* connectome. The thickness of these arrows indicate the number of synapses at a given connection. Dotted blue and orange arrows indicate neuromodulatory connections from ***Flavell et al., 2013***. (**C**) Example dataset from multi-neuron calcium imaging in a free-moving wild-type animal. The calcium activity of each neuron is shown in black. The green-red heat map in the background indicates axial velocity of the animal, and the behavioral state of the animal is shown on top. GCaMP6m data were divided by co-expressed mScarlett fluorescence levels and normalized to a 0–1 scale, based on the 1st and 99th percentiles of the neuron's signal (see Materials and methods). (**D**) Event-triggered averages of individual neuron activity aligned to transitions between roaming ('R') and dwelling ('D') (left column), or transitions between forward runs ('F') and reversals ('RV') (right column). Data are shown as means and 95 % confidence interval (95% CI). (**E**) Histograms of individual neuron's activity during roaming (orange), dwelling (blue), forward runs (green), or reversals (red). Note that shifts of distributions to the right indicate increased neural activity. (**F**) Simultaneously recorded activity of the 10 neurons projected onto the space spanned by the 1st and 2nd principal components (i.e. PC1 and PC2). Individual data points are colored according to the ongoing axial velocity. Histograms above and to the right of the scatterplot indicate distribution of PC1 and PC2 values for 3 ranges of axial velocity (*v*): $v \leq -0.0165$ mm/s (red), $-0.0165$ mm/s $< v \leq 0.0165$ mm/s (gray), $v \geq 0.0165$ mm/s (green). (**G**) Projection of neural activity in principal component space, colored by the ongoing foraging state. Histograms show distributions of PC1 or PC2 values conditioned on the foraging state. (**H**) Comparison of measured velocity (x-axis) to the velocity predicted by a General Linear Model that was trained from the neural data (y-axis). The density of datapoints in this space is represented as a two-dimensional histogram. (**I**) Average Receiver Operating Curves from logistic regression models trained to predict foraging states using ongoing neural activity data from all 10 neurons or subsets of neurons (see Supplemental Methods for details). Dotted line indicates level expected by chance. Data in D-H are from the same set of wild-type animals (N = 17). *p < 0.05, **p < 0.01, ***p < 0.001, ****p < 0.0001, Wilcoxon rank-sum test.

The online version of this article includes the following figure supplement(s) for figure 1:

**Figure supplement 1.** Design and calibration of the spinning-disk confocal tracking scope.

**Figure supplement 2.** Calibration of behavioral tracking accuracy and the effect of motion on calcium imaging data.

**Figure supplement 3.** Additional examples of multi-neuron calcium activity traces in freely moving wild-type animals.

**Figure supplement 4.** Gaussian Mixture Models (GMM) for analyzing animal speed and NSM calcium activity.

**Figure supplement 5.** Encoding of behavioral parameters by the calcium activity of individual neurons.

**Figure supplement 6.** Differences in RIB and AVA joint activity during roaming compared to dwelling.

**Figure supplement 7.** Relationships between specific neurons and the principal components.

**Figure supplement 8.** Examples showing the prediction of locomotion parameters using circuit activity.

activities of multiple neurons, including NSM and a set of neurons that have previously been shown to control forward and reverse locomotion.

The encoding of locomotion and behavioral state across many neurons suggests a circuit-level representation of the animal's behavior. To test whether the dominant modes of activity in the circuit were associated with the animal's behavior, we performed Principal Component Analysis (PCA) using the activity profiles of all the recorded neurons. Indeed, the top two principal components (PC1 and PC2), which together explained 44% of the total variance, exhibited clear behavioral correlates (***Figure 1F–G***). Neural activity along PC1 was coupled to the animal's forward and reverse locomotion (***Figure 1F***), while activity along PC2 was coupled to the animal's axial speed and foraging state (***Figure 1G***). PC1 consisted of positive contributions from forward-run-active neurons (e.g. AVB, AIY, RIB, and RME), a negative contribution from the reversal-active neuron AVA, and almost no contribution from the serotonergic neuron NSM (***Figure 1—figure supplement 7***). These results provide a clear match to our single-neuron analyses above and suggest that PC1 primarily encodes movement direction. In contrast, PC2 consisted of a strong positive contribution from NSM, which is active at low speeds, and negative contributions from both forward-run- and reversal-active neurons (***Figure 1— figure supplement 7***). These results are also consistent with our single neuron analyses and suggest that PC2, which primarily encodes dwelling, is associated with elevated NSM activity and decreased activity in the forward-run- and reversal-active neurons. These results indicate that the main modes of dynamics in this circuit are associated with the animal's movement direction and foraging state.

Finally, to test whether neural activity in the roaming-dwelling circuit was sufficient to accurately predict behavior, we trained statistical models to predict animal velocity and behavioral state from neural activity (***Figure 1H–I***; ***Figure 1—figure supplement 8***). A nonlinear regression model was able to predict animal velocity from concurrent neural activity data with a high degree of accuracy (***Figure 1H***). This observation provides a rough match to a previous study (***Hallinen et al., 2021***) and is consistent with the known roles of several of these neurons in controlling velocity (***Kato et al., 2015***;

*Wang, 2020*; *Roberts et al., 2016*; *Li et al., 2014*; *Luo et al., 2014*; *Chalfie et al., 1985*). In addition, a logistic regression model trained to predict the roaming and dwelling state achieved over 95 % accuracy when using activity from all 10 neurons, and exhibited classification accuracy significantly above baseline when using data from only NSM or the roaming-active neurons AIY, RIB, and AVB (*Figure 1I*). Thus, ongoing neural activity in the roaming-dwelling circuit can predict the animal's locomotion and foraging state. This robust mapping between circuit activity and behavior raised the possibility that stable activation of one or more neurons in this circuit might underlie persistent roaming and dwelling states.

## Persistent NSM activation and associated circuit activity changes correspond to the dwelling state

Among the neurons recorded, the serotonergic neuron NSM was unique in that its persistent activation was closely aligned to the dwelling state (*Figure 1D–E*; *Figure 1—figure supplement 5*). NSM activity was increased during dwelling as compared to roaming, resulting in a negative correlation with animal speed, which differs across the two states (*Figure 2A*). However, NSM activity was not correlated with speed within the roaming state and it was not associated with movement direction during any state (*Figures 2A and 1D–E*). An increase in NSM activity consistently preceded dwelling, with an average latency of 23 s (*Figure 2B–C*). Dwelling states frequently ended with a decrease in NSM activity, though with a more variable latency (*Figure 2C* and *Figure 2—figure supplement 1A*). Across wild-type animals, the durations of individual dwelling states were positively correlated with the durations of co-occurring bouts of NSM activity, which both typically lasted many minutes (*Figure 2D*). Together, these observations indicate that NSM displays persistent dwelling state-associated neural activity.

Since previous work showed that optogenetic NSM activation can drive animals into dwelling states (*Rhoades et al., 2019*; *Flavell et al., 2013*), our observation that NSM is persistently active during dwelling raised the possibility that NSM activation may play an important role in organizing the circuit-wide activity changes that accompany dwelling. To further explore this possibility, we examined how circuit activity evolved in PC space during periods of NSM activation (*Figure 2E–F*). NSM activity serves as a major component of PC2 and is only weakly represented on PC1 (*Figure 1—figure supplement 7*). As a result, high and low NSM activity segregates well along PC2, but not PC1 (*Figure 2E*; *Figure 2—figure supplement 1B*). By aligning circuit activity to the onset of NSM activity bouts and projecting the activity in PC space, we found that NSM activation often began when circuit activity was in the region of the PC space with high PC1 activity and low PC2 activity, typical of forward locomotion during roaming (*Figure 2F*, compare with *Figure 1F*). As NSM became active, circuit activity rose rapidly along PC2 and stayed within the positive half of PC2 (a region typically associated with low speed and dwelling; see *Figure 1F–G*). Afterwards, circuit activity slowly traveled towards low values of both PC1 and PC2 (a region typically associated with reversals; see *Figure 1F*). This observation suggests that the persistent activation of NSM during dwelling is associated with stereotyped changes in overall circuit dynamics. To test whether NSM activation was sufficient to drive these changes in circuit dynamics, we stimulated NSM using the red-shifted opsin Chrimson while imaging circuit activity. Indeed, optogenetic stimulation of NSM (performed at a low all-trans-retinal (ATR) concentration to avoid background activation; see Materials and methods) inhibited the activity of the roaming-active neurons AVB, AIY, and RIB and led to a decrease in animal speed (*Figure 2G*). These effects were largely abolished in mutants lacking the 5-HT-gated chloride channel *mod-1* (*Figure 2—figure supplement 1C*), consistent with previous reports that the *mod-1* receptor is critical for 5-HT-induced locomotion changes (*Flavell et al., 2013*; *Sawin et al., 2000*; *Iwanir et al., 2016*). Taken together, these results indicate that NSM activation is associated with and can drive, at least in part, stereotyped changes in circuit activity characteristic of the dwelling state.

## Persistent activity in serotonergic NSM neurons requires feedback from its target neurons that express the MOD-1 serotonin receptor

Our analyses of wild-type circuit dynamics revealed that stereotyped changes in circuit activity are associated with the roaming and dwelling states. We next examined how these neural dynamics are influenced by neuromodulatory connections embedded in the circuit. Although the 5-HT and PDF systems are known to act in opposition to regulate roaming and dwelling behaviors (*Flavell et al.,*

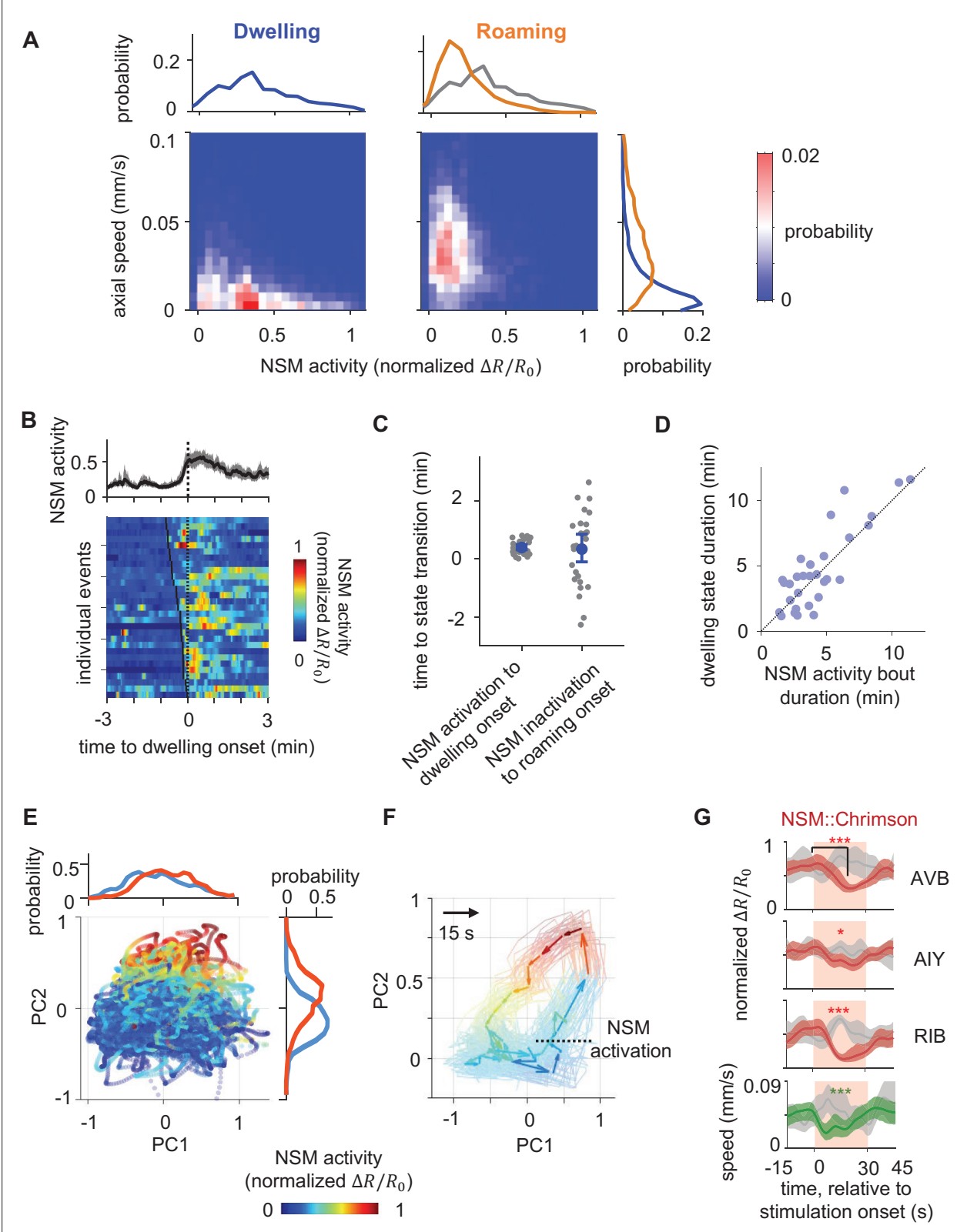

**Figure 2.** Persistent NSM activity is associated with the dwelling state. (**A**) Joint distribution of NSM activity and the concurrent axial velocity during the dwelling (left column) or the roaming (right column) state. Histograms on top show marginal distributions of NSM activity during dwelling (left) or roaming (right). Histogram to the right show marginal distributions of axial velocity during dwelling (blue) or roaming (orange) states. (**B**) NSM activity aligned to the onset of dwelling states. (Top) Average NSM activity around the onset of dwelling states. (Bottom) Heat map of NSM activity around

*Figure 2 continued on next page*

*Figure 2 continued*

individual instances of roaming-to-dwelling transitions. Dotted black line denotes the onsets of dwelling states. Black ticks on the heat map mark the onset of an NSM activity bout. (**C**) Left: latencies of roaming-to-dwelling transitions relative to the closest onset of an NSM activity bout. Right: latencies of dwelling-to-roaming transitions relative to the closest offset of an NSM activity bout. NSM activity bouts are defined through Gaussian Mixture Clustering (see Methods for details). (**D**) Scatterplot of the durations of individual dwelling states and the durations of their coinciding NSM activity bouts. (**E**) Projection of neural activity in principal component space, colored by concurrent NSM activity. Histograms show distributions of NSM activity along both axes. (**F**) Average circuit activity dynamics in principal component space aligned to the onset of NSM activation. Each colored arrow represents average activity dynamics over a 15 s interval. Color indicates ongoing NSM activity. Faint lines show bootstrap samples of the average dynamics. (**G**) Event triggered averages of individual neuron activity and animal speed aligned to the optogenetic activation of NSM. Red and green traces represent data from animals raised on all-trans retinal (ATR) (N = 6 animals), while gray traces represent data from control animals raised without ATR (N = 4 animals). Light red patch indicates the time window in which the red light is turned on. Comparisons are made between 1 s before the onset of the red light stimulation and 18 s into the stimulation. Data are shown as means and 95% C.I.s **p < 0.01, ***p < 0.001, ****p < 0.0001, Wilcoxon rank-sum test with Benjamini-Hochberg (BH) correction.

The online version of this article includes the following figure supplement(s) for figure 2:

**Figure supplement 1.** Further analysis of NSM activity.

*2013*), it is not known how these neuromodulators impact circuit dynamics. To address this question, we imaged neural activity in mutants deficient in 5-HT signaling, PDF signaling, or both (*Figures 3 and 4*). Mutants that disrupt 5-HT signaling, such as those lacking a key enzyme for serotonin biosynthesis (*tph-1*) or a 5-HT-gated chloride channel (*mod-1*), exhibited a decrease in time spent in the dwelling state (*Figure 3A–B*, *Figure 3—figure supplement 1*, *Figure 3—figure supplement 2*), consistent with previous results (*Flavell et al., 2013*). In wild-type animals, NSM activity was strongly associated with reduced speed, but this relationship was attenuated in *tph-1* and *mod-1* mutants (*Figure 3C*). Surprisingly, we also found that the durations of the NSM activity bouts, which were minutes-long in wild-type animals, were dramatically shortened in these mutants (*Figure 3D*). This resulted in a significant decrease in the fraction of time that NSM is active in the mutants (*Figure 3E*). These results indicate that 5-HT signaling is required to sustain the activity of the serotonergic neuron NSM. Because MOD-1 is an inhibitory 5-HT-gated chloride channel, the *mod-1*-expressing neurons are relieved from inhibition by 5-HT in *mod-1* mutants. Thus, the decrease in NSM activity in these mutants suggests an inhibitory role for the *mod-1*-expressing neurons in regulating NSM activity. Previous work has shown that *mod-1* functions in the neurons AIY, RIF, and ASI to promote dwelling (*Flavell et al., 2013*; *Figure 1B*). Since none of these neurons directly synapse onto NSM, they must functionally inhibit NSM through a polysynaptic route or via the release of a neuromodulator. To directly test whether activation of these neurons inhibits NSM, we activated the *mod-1*-expressing neurons with Chrimson while recording NSM activity. We delivered the optogenetic stimuli specifically when NSM activity was high and observed a sustained inhibition of NSM activity throughout the stimulation (*Figure 3F*). Together, these results indicate that the serotonergic NSM neuron promotes its own activity via mutual inhibition with neurons expressing the inhibitory 5-HT receptor MOD-1 (*Figure 3G*).

## PDF receptor-expressing neurons inhibit NSM to promote mutual exclusivity between NSM and AVB

We next examined the impact of PDF signaling on circuit dynamics by imaging animals carrying a null mutation in the PDF receptor gene *pdfr-1* (*Figure 4A* and *Figure 4—figure supplement 1A*). In wild-type animals, the serotonergic neuron NSM and the PDF-1-producing neuron AVB exhibited a mutually exclusive activity pattern, wherein NSM activity was high and AVB activity was low during dwelling, while NSM activity was low and AVB was dynamically active during roaming (*Figure 4C*; here we define 'mutual exclusivity' to be a lack of concurrent high activity in NSM and AVB; see Materials and methods and *Figure 4—figure supplement 2A* for thresholding approach to segment low versus high activity). This mutual exclusivity was strongly disrupted in *pdfr-1* mutants (*Figure 4C–D*). In addition, an analysis of graded AVB activity changes during periods of NSM activation confirmed that the overall decrease in AVB activity during periods of high NSM activity was disrupted in *pdfr-1* mutants (*Figure 4—figure supplement 2*). In these mutant animals, the two neurons were frequently co-active, giving rise to a positive correlation between the activities of the two neurons (*Figure 4D* and *Figure 4—figure supplement 3*). Positive correlations also appeared between NSM and other roaming-active neurons, including the *pdfr-1*-expressing neurons AIY and RIB (*Figure 4—figure*

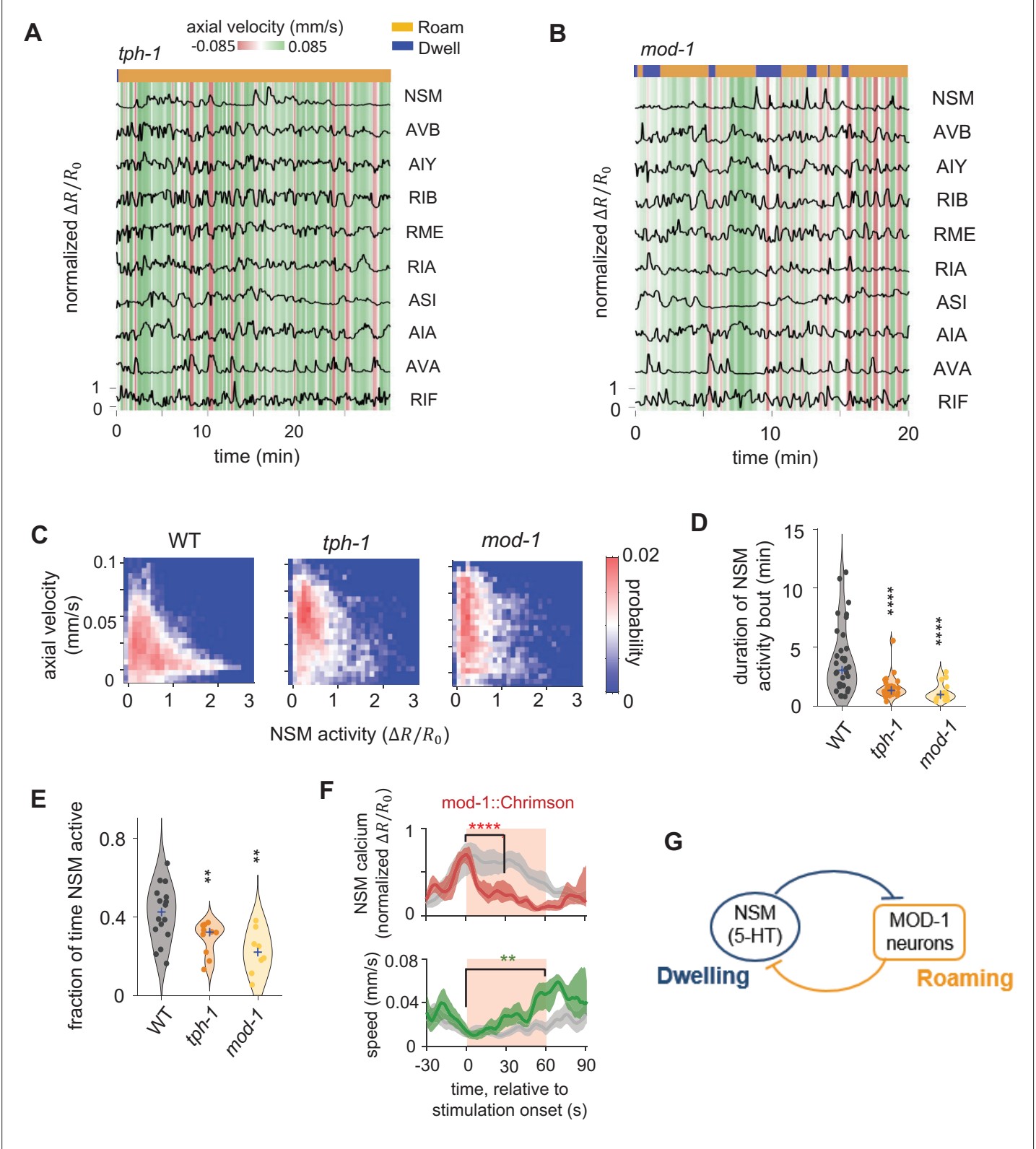

**Figure 3.** Serotonin signaling promotes persistent activation of serotonergic NSM neurons via a mutual inhibitory circuit. (**A–B**) Example circuit-wide calcium imaging datasets from *tph-1* (**A**) and *mod-1* (**B**) mutant animals, shown as in *Figure 1C*. (**C**) Association of NSM activity and axial speed in the indicated genotypes. Data are shown as probability density plots. (**D**) Duration of NSM activity bouts for the indicated genotypes. Data points represent individual NSM activity bouts and violin plots show distributions across animals of the same genotype. Blue '+' marks the median of each distribution.

*Figure 3 continued on next page*

*Figure 3 continued*

(**E**) Probability of NSM being active in wild type animals and serotonin mutants. Data points represent individual animals and violin plots show distributions across animals of the same genotype. Blue '+' marks the median of each distribution. For (**D–E**), N = 17, 10, and 8 animals for WT, *tph-1*, and *mod-1*, respectively. (**F**) Event triggered averages of NSM activity and animal speed aligned to the optogenetic activation of *mod-1* expressing neurons. Red and green traces represent data from animals raised on all-trans retinal (ATR) (N = 7 animals), while gray traces represent data from control animals raised without ATR (N = 4 animals). Data are shown as means and 95% C.I.s. Light red patch indicates the time window in which the red light is turned on. For NSM calcium activity, the comparison is made between 1 s before the onset of the red light stimulation and 30 s into the stimulation. For animal speed, the comparisons is made between 1 s before the onset of the red light stimulation and 60 s into the stimulation. (**G**) Circuit schematic based on results from the *tph-1* and *mod-1* mutants, showing cross inhibition between the NSM and the MOD-1 expressing neurons. For (**D–F**), \*\*p < 0.01, \*\*\*p < 0.001, \*\*\*\*p < 0.0001, Wilcoxon rank-sum test with Benjamini-Hochberg (BH) correction.

The online version of this article includes the following figure supplement(s) for figure 3:

**Figure supplement 1.** Further analysis of NSM activity.

**Figure supplement 2.** Additional examples of multi-neuron calcium activity traces in free-moving serotonin mutants.

*supplement 3*). This increased co-activity of NSM and AVB was observed when using multiple distinct GCaMP normalization methods and when sampling from matched speed distributions in wild-type and mutant animals (*Figure 4—figure supplement 4*). We observed that *pdfr-1* animals frequently moved at speeds mid-way between those typically seen for roaming and dwelling states in wild-type animals (*Figure 4—figure supplement 5A-B*; this observation prompted us to not perform roaming-dwelling state calls on the *pdfr-1* mutant). One likely explanation for this behavioral phenotype is that ectopic co-activation of NSM and the roaming-active neurons results in mixed behavioral outputs that differ from either roaming or dwelling. These findings indicate that PDF signaling is required for the neural circuit to maintain mutual exclusivity between NSM and the locomotion-controlling neurons that are active during roaming.

In contrast to the *tph-1* animals, NSM activity bouts in *pdfr-1* mutants were more persistent than they were in wild-type animals (*Figure 4E–F*). This result suggests that PDF signaling plays an important role in suppressing NSM activity. Consistent with this interpretation, constitutive activation of PDFR-1 signaling, via expression of the hyperactive PDFR-1 effector ACY-1(P260S) in the *pdfr-1*-expressing neurons, strongly inhibited NSM activity (*Figure 4B*; *Figure 4E–F*; *Figure 4—figure supplement 4B*). In addition, optogenetic activation of the *pdf-1*-expressing neurons led to an acute and robust inhibition of NSM (*Figure 4G*; *Figure 4—figure supplement 5C*). Together, these findings indicate that PDF signaling is necessary and sufficient to keep NSM inactive during roaming, a key requirement for generating the opposing roaming and dwelling states.

## PDFR-1 neurons act downstream of MOD-1 neurons to inhibit NSM activity and promote roaming

To probe whether the MOD-1- and PDFR-1-expressing neurons act in the same pathway to impact NSM activity, we performed epistasis analysis by examining *tph-1;pdfr-1* double mutants. Similar to the *pdfr-1* single mutants, these animals exhibited prolonged bouts of NSM activation, an increased probability of NSM being active, and a near two-fold increase in the probability of co-activation between NSM and AVB (*Figure 4C–F*). At the behavioral level, *tph-1;pdfr-1* animals spent over a third of their time moving at intermediate speeds, similar to the *pdfr-1* animals (*Figure 4—figure supplement 5A-B*). Together, these results suggest that *pdfr-1* functions downstream of *tph-1* to control NSM activity and locomotion.

It has been shown that 5-HT targets the *mod-1*-expressing neurons to inhibit roaming while *pdfr-1* functions in multiple *pdfr-1*-expressing neurons, including RIM, AIY, RIA, and RIB, to promote roaming (*Rhoades et al., 2019*). To test whether *mod-1*-expressing neurons and *pdfr-1*-expressing neurons act in the same neuronal pathway to control foraging states, we optogenetically activated *mod-1*-expressing neurons in either wild-type animals or *pdfr-1* mutants. We found that optogenetic activation of the *mod-1*-expressing neurons, which triggered high-speed locomotion in wild-type animals, failed to do so in *pdfr-1* mutants (*Figure 4H*). These results indicate that the *pdfr-1*-expressing neurons act downstream of the *mod-1*-expressing neurons to promote roaming (*Figure 4I*), consistent with the epistasis analysis described above. Altogether, these results indicate that the mutually inhibitory interaction between NSM and the neurons that express the MOD-1 and PDFR-1 receptors is necessary to stabilize the opposing neural activity and behavioral patterns underlying roaming and dwelling.

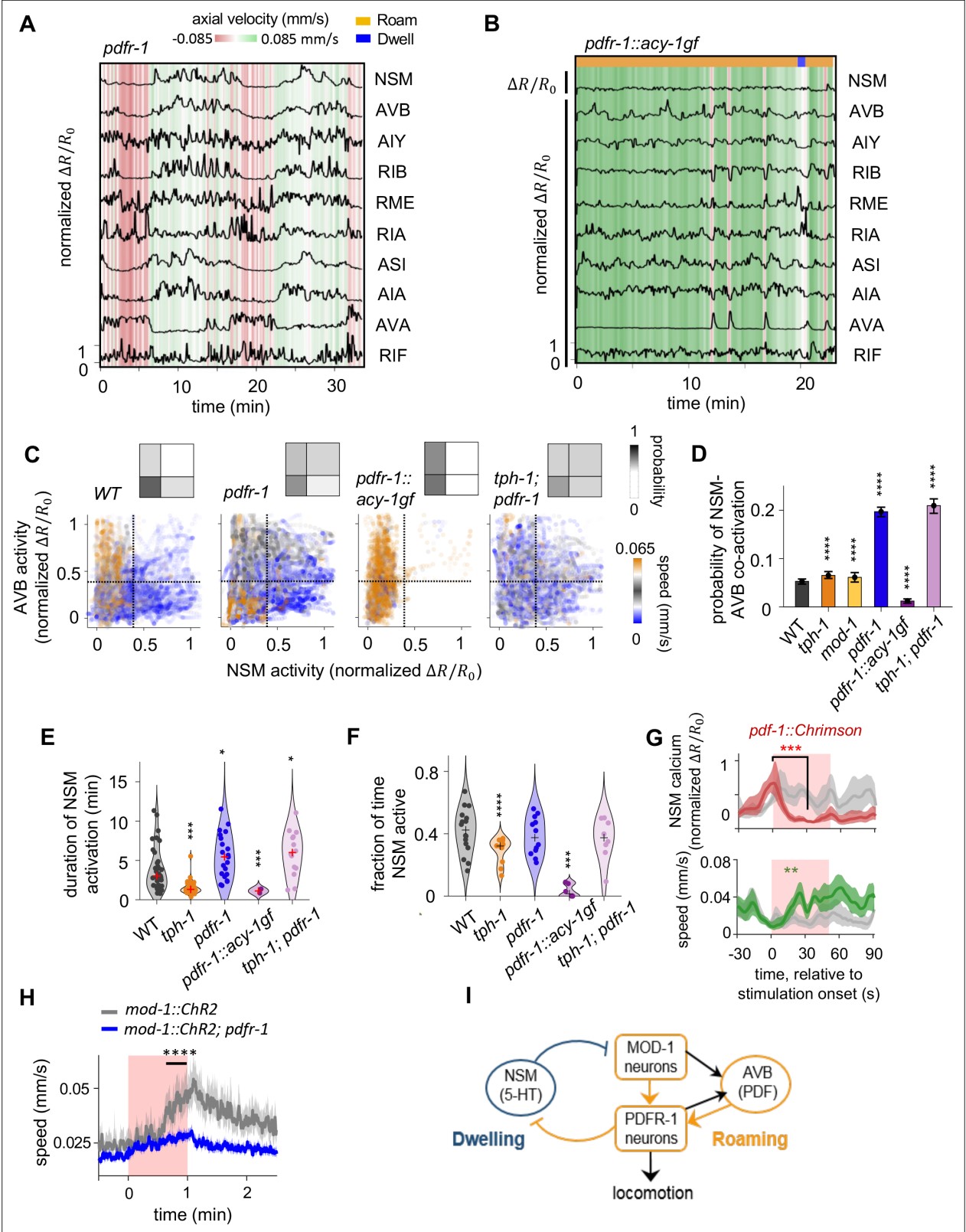

**Figure 4.** PDF signaling is required for mutual exclusivity between circuit states and acts downstream of the 5-HT target neurons in the mutual inhibitory circuit. (**A**) Example circuit-wide calcium imaging dataset from *pdfr-1* mutants lacking PDF neuropeptide signaling, shown as in *Figure 1C*. No roaming/dwelling ethogram is shown for *pdfr-1* animals due to changes in their speed distribution that implicate altered or new behavioral states (see *Figure 4—figure supplement 4A-B*). (**B**) Example circuit-wide calcium imaging dataset from transgenic animals expressing the hyperactive PDFR-1

*Figure 4 continued on next page*

*Figure 4 continued*

effector ACY-1(P260S) specifically in *pdfr-1* expressing neurons. For NSM, the un-normalized $R/R_0$ is shown since the $R/R_0$ values never exceeded 10 % of the average peak NSM activity wild-type animals. (**C**) Scatterplots of NSM and AVB activity in *pdfr-1* mutants, transgenic *pdfr-1::acy-1(P260S) gf* animals, and *tph-1; pdfr-1* double mutants. Data points are colored by the instantaneous speed of the animal. Color scale was chosen so that blue colors correspond to speeds typical of the dwelling state, orange correspond to speeds typical of the roaming states, while gray colors indicate speeds in-between the former. Dotted lines show the threshold values for NSM and AVB activity used for defining 'co-activity' (determined using the Otsu method; see Methods). Insets show the density of data points in each of the quadrants defined by these threshold activity levels. (**D**) Probability of NSM and AVB being co-active for genotypes shown in (**C**). ***p < 0.001; ****p < 0.0001, bootstrap estimates of the mean with BH correction. (**E**) Duration of NSM activity bouts for the indicated genotypes. Data points corresponds to individual NSM activity bouts. Each violin plot represent data from animals of the same genotype. '+' denotes the median of each distribution. (**F**) Probability of NSM being active in wild-type and mutant animals. Data points corresponds to individual NSM activity bouts. Each violin plot represent data from animals of the same genotype. '+' denotes the median of each distribution. For (**C–F**), N = 17, 10, 8, 11, 9, and 8 animals for WT, *tph-1*, and *mod-1*, *pdfr-1*, *pdfr-1::acy-1gf*, and *tph-1;pdfr-1* animals. (**G**) Event triggered averages of NSM activity and animal speed aligned to the optogenetic activation of *pdf-1* expressing neurons. Red and green traces represent data from animals raised on all-trans retinal (ATR) (N = 5 animals), while gray traces represent data from control animals raised without ATR (N = 4 animals). Data are shown as means and 95% C.I.s. For NSM calcium activity and animal speed, comparisons are made between 1 s before the onset of the red light stimulation and 30 s into the stimulation. (**H**) Speed of wild-type and *pdfr-1* mutant animals in response to optogenetic activation of the MOD-1 expressing neurons (red shading). Average speeds during the window spanned by the black line were compared between animals of the two genotypes. (**I**) Circuit schematic summarizing results shown in (**C–H**): the PDFR-1 expressing neurons act downstream of the MOD-1 expressing neurons to inhibit the 5-HT neuron NSM. Black arrows indicate anatomical connections based on the *C. elegans* connectome (*White et al., 1986*). For (**D- H**), **p < 0.01, ***p < 0.001, ****p < 0.0001, Wilcoxon rank-sum test with BH correction.

The online version of this article includes the following figure supplement(s) for figure 4:

**Figure supplement 1.** Additional examples of multi-neuron calcium activity traces in free-moving serotonin mutants.

**Figure supplement 2.** Further analysis of NSM-AVB co-activity.

**Figure supplement 3.** Correlations between neurons in wild-type and mutant animals.

**Figure supplement 4.** Joint activity of NSM and AVB in serotonin and PDF signaling mutants.

**Figure supplement 5.** Analyses of circuit dynamics and foraging behavior in serotonin and PDF signaling mutants.

## A CNN classifier reveals stereotyped circuit dynamics that precede roaming-to-dwelling transitions

Based on the *C. elegans* connectome and previous studies (*Li et al., 2014*; *Luo et al., 2014*; *Gordus et al., 2015*; *Iino and Yoshida, 2009*; *Hendricks et al., 2012*), many of the MOD-1- and PDFR-1-expressing neurons receive dense inputs from sensory neurons and are functionally involved in sensorimotor behaviors (*Figure 1B*). Therefore, the functional circuit architecture revealed through our calcium imaging analyses raised the possibility that incoming sensory inputs that act on the MOD-1- and PDFR-1-expressing neurons might influence the transitions between roaming and dwelling. One prediction of this hypothesis is that these neurons that receive sensory inputs may display stereotyped activity patterns prior to state transitions.

To test the above hypothesis, we sought to predict state transitions from the circuit-wide activity patterns that precede them. Our calcium imaging results showed that the onset of NSM activity reliably coincided with the onset of dwelling states (*Figure 2B and C*). We thus focused on uncovering potential circuit elements that function upstream of NSM to drive the roaming-to-dwelling transition. We adopted a supervised machine learning approach by training a Convolutional Neural Network (CNN) classifier to predict NSM activation using the preceding multi-dimensional activity profile from all other neurons imaged (*Figure 5A–B* and *Figure 5—figure supplement 1*; see Materials and methods). We chose the CNN classifier because of its flexible architecture, which can model complex nonlinear relationships between the input and output variables and detect multiple relevant activity patterns via the same network (*Maheswaranathan et al., 2017*; *Maheswaranathan et al., 2018*; *McIntosh et al., 2016*). Successfully trained networks achieved over 70 % test accuracy, equaling or exceeding other supervised learning methods (*Figure 5—figure supplement 2A*). This result indicates that stereotyped circuit activity patterns frequently precede NSM activation.

We examined the parameters of the trained networks to define the activity patterns that were being used to make successful predictions about upcoming NSM activation. Successfully trained networks consistently employed a convolutional filter where the largest positive weights were associated with the sensory processing neuron AIA and the largest negative weights were linked to RIB and AVA, which promote forward and reverse movement, respectively (*Figure 5B*). These weights

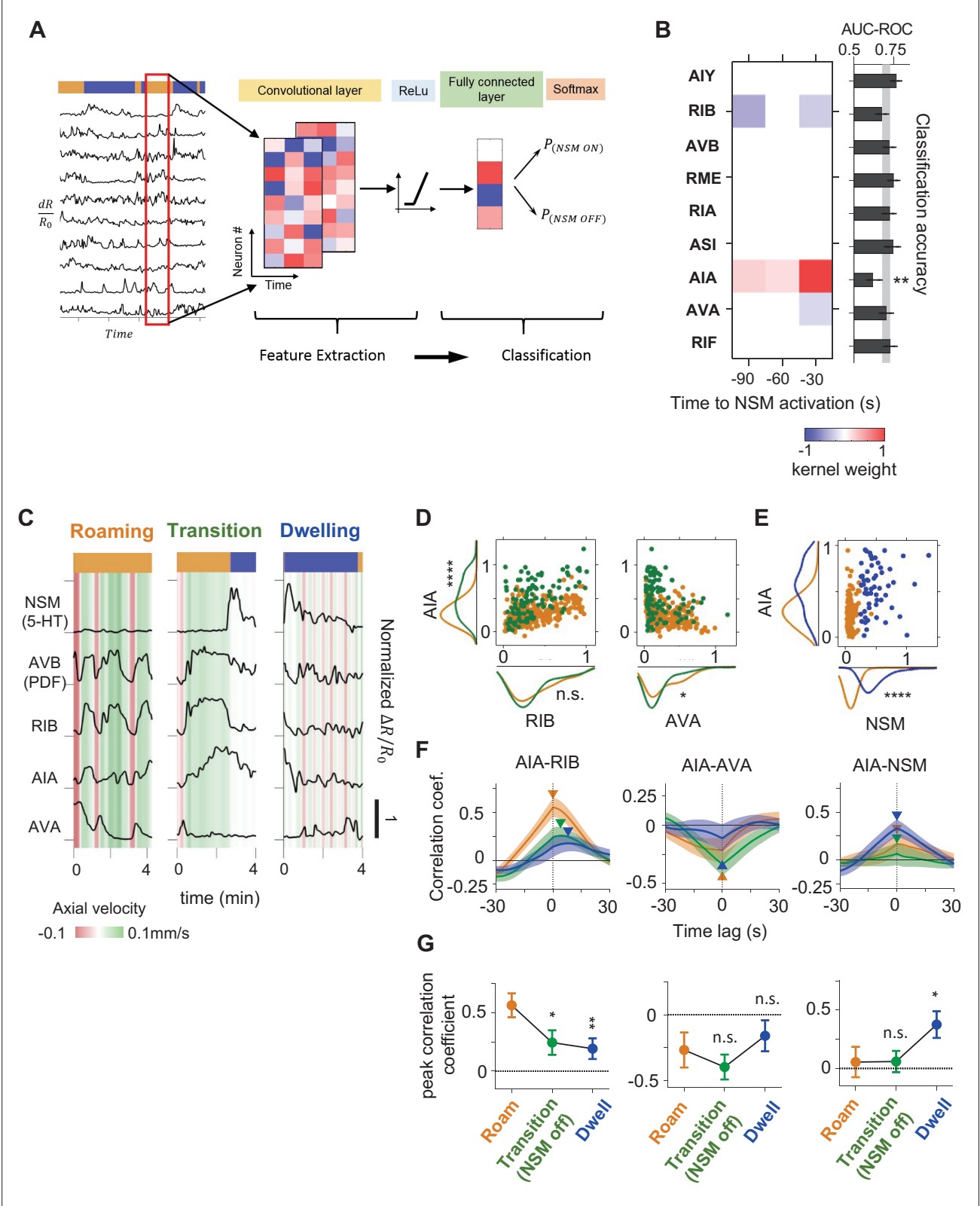

**Figure 5.** A CNN classifier identifies circuit activity patterns predictive of roaming-to-dwelling state transitions. (**A**) Schematic illustrating the architecture of the Convolutional Neural Network (CNN) trained to predict NSM activation events. (**B**) Left: a common convolutional kernel found across successfully trained CNNs. Only weights that are significantly different from zero are colored. Right: Feature selection results. Each black bar depicts the average area under the curve for the Receiver Operating Characteristic curve (AUC-ROC) from networks trained using data with one neuron held out at a time.

*Figure 5 continued on next page*

*Figure 5 continued*

The identity of the held-out neuron is indicated to the far left. The gray stripe in the background denotes the 95% CI of the AUC-ROC from networks trained using data from all nine neurons. Error bars are 95% CI of the mean. **p < 0.01, bootstrap estimate of the mean with BH correction. (**C**) Example activity traces from NSM, AVB, and the three neurons with significant weights in the convolutional kernel. Activity traces were taken during roaming (left), dwelling (right) and roaming-to-dwelling transition. (**D**) Scatterplots of simultaneously measured neural activity of the indicated pairs of neurons. Orange data points are taken during roaming states at least 1 min before the onset of the next dwelling states and before NSM becomes active. Green data points are taken within 1 min before the onset of dwelling states. Along the x- and the y- axes are marginal probability distributions of the data points shown in the scatterplots. (**E**) Scatterplots of simultaneously measured neural activities of AIA and NSM. Orange data points are taken within 1 min before the onset of the next dwelling states and before NSM becomes active. Blue data points are taken within 30 s after the onset of dwelling states. Along the x- and the y- axes are marginal probability distributions of the data points shown in the scatterplots. (**F**) Average cross-correlation functions between the indicated pairs of neurons during roaming (orange, data taken from 100 to 70 s before the onset of the next dwelling state), roaming-to-dwelling transition (green, data taken from 30 to 0 s before the NSM activation event prior to the onset of the next dwelling state), or dwelling (blue, data taken from 10 to 40 s after dwelling onset). Error bars are standard error of the mean. Arrowheads denote the point of maximum in absolute magnitude of the cross-correlation function. (**G**) Average cross-correlation coefficients computed at peak points indicated in (**F**). For (**B, D–G**), N = 17 WT animals. For (**D–E and G**), *p < 0.05, **p < 0.01, ****p < 0.0001, Wilcoxon rank-sum test with BH correction.

The online version of this article includes the following figure supplement(s) for figure 5:

**Figure supplement 1.** Parameter selection for the CNN model.

**Figure supplement 2.** Evaluation of CNN classifier performance.

**Figure supplement 3.** Convolutional kernels trained to predict transitions in foraging state or NSM activity.

suggest that NSM activation is most likely to occur following increased activity in AIA and decreased activity in RIB and AVA. Withholding AIA, RIB, and AVA from the training data abolished the predictive power of the trained network, while withholding AIA activity alone also led to a significant reduction in test accuracy (*Figure 5B* and *Figure 5—figure supplement 2B*). Moreover, networks trained on the activities of only AIA, RIB, and AVA performed nearly as well as those trained on all the neurons (*Figure 5—figure supplement 2B*). Training a CNN classifier to directly predict dwelling state onset from all of the neurons except NSM led to a similar convolutional filter; including NSM in the training data yielded a classifier that predicts dwelling state onset solely using NSM activity (*Figure 5—figure supplement 3A*). These observations suggest that the combined activities of AIA, RIB, and AVA can frequently predict the onset of NSM activity, which is strongly associated with roaming-to-dwelling state transitions.

Utilizing this same approach, we also trained CNN classifiers to predict the termination of NSM activity bouts and the onsets of roaming states (*Figure 5—figure supplement 3B-C*). The resulting convolutional kernels displayed strong positive weight on AVB and RIB. Strong negative weight on NSM was also a feature of the convolutional kernel predicting roaming state onset. These results are consistent with the mutual inhibitory loop described above and suggest that activation of the AVB and other roaming-active neurons, concurrent with NSM inactivation, predicts dwelling-to-roaming transitions.

Given that AIA, RIB, and AVA activities could predict the onset of NSM activity and dwelling states, we next examined how the activities of these neurons changed during transitions from roaming to dwelling (*Figure 5C–G*). During roaming, AIA activity was positively correlated with that of forward run-promoting neurons, such as RIB, and negatively correlated with the reversal-promoting neuron AVA. Within 30 s of NSM activation, AIA often exhibited a further increase in activity, while RIB and AVA activity stayed at similar levels or decreased. As NSM activity rose and the animal entered the dwelling state, RIB and AVA activity declined sharply while AIA became correlated with NSM. AIA then declined to baseline over the following minutes. Thus, AIA activity co-varies with the forward-active neurons during roaming and with NSM at the onset of dwelling. This native activity pattern is consistent with the convolutional kernel from the CNN classifier, where heightened activity of AIA relative to the locomotion-promoting neurons RIB and AVA predicts NSM activation. Together, these results reveal a stereotyped, multi-neuron activity pattern that predicts NSM activation.

## AIA activation can elicit both roaming and dwelling states

Because AIA activity co-varied with both roaming- and dwelling-active neurons and was required for the prediction of NSM activation, we hypothesized that AIA might play an active role in controlling the transitions between roaming and dwelling. To test this, we optogenetically activated AIA in foraging

animals exposed to uniform lawns of bacterial food (*Figure 6A–B*). Behavioral responses to AIA activation depended on the state of the animal at the time that AIA was activated. Roaming animals exhibited a rapid and transient decrease in speed upon AIA activation, while dwelling animals showed a gradual increase in speed upon AIA activation (*Figure 6A–B*). These results indicate that optogenetic activation of AIA can affect state transitions on two different time scales: triggering the roaming-to-dwelling transition within a few seconds and promoting entry into the roaming state upon tens of seconds of continued activation.

To determine if AIA promotes behavioral switching by modulating 5-HT- or PDF-releasing neurons, we optogenetically activated AIA in mutants defective in 5-HT or PDF signaling (*tph-1* and *pdfr-1* animals, *Figure 6A–B*). For the *tph-1* mutant, animals that were roaming pre-stimulation no longer displayed rapid entry into dwelling and showed a higher probability of staying in the roaming state later into the stimulation. *tph-1* animals that were dwelling pre-stimulation displayed a higher probability of entering roaming compared to control animals. These results suggest that *tph-1* is critical for the effect of AIA activation on triggering entry into dwelling and for preventing AIA-induced entry into the roaming state. In contrast, AIA activation in *pdfr-1* mutants that were dwelling pre-stimulation failed to promote transitions into roaming. Roaming states in these mutants were too infrequent and brief to warrant meaningful analysis of AIA activation during that state. Together, these results suggest that AIA promotes dwelling via 5-HT signaling and promotes roaming via PDF signaling.

Previous work has characterized neuronal cell types in mammals that exhibit similar trial-by-trial variation where optogenetic activation can elicit opposing behavioral effects (*Seo et al., 2019*; *Lee et al., 2014*). In some of these previous examples, stimulation intensity influenced the behavioral outcome of the optogenetic activation. Thus, we examined whether stimulation intensity influenced the ability of AIA to promote roaming or dwelling. Indeed, AIA activation at lower light intensities primarily elicited roaming-to-dwelling transitions, while activation at higher intensities elicited dwelling-to-roaming transitions (*Figure 6C*). Because the AIA-induced slowing response and speeding response depend on different neuromodulatory systems and can be elicited independently at different light intensities, these results are suggestive that AIA provides independent outputs to the PDF and 5-HT systems to control roaming and dwelling states, respectively (*Figure 6D*).

## AIA can promote either roaming or dwelling, depending on the sensory context

Based on the *C. elegans* connectome (*Cook et al., 2019*; *White et al., 1986*), AIA receives the majority of its synaptic inputs (~80%) from chemosensory neurons (*Figure 6—figure supplement 1*), many of which detect temporal changes in the concentrations of olfactory and gustatory cues (*Chalasani et al., 2007*; *Larsch et al., 2015*; *Suzuki et al., 2008*). Previous work has shown that AIA is activated by an increase in the concentration of attractive odorants present in bacterial food (*Larsch et al., 2015*; *Dobosiewicz et al., 2019*). In the absence of food, AIA promotes forward runs when animals detect increases in attractive odors (*Larsch et al., 2015*). AIA sends synaptic output to multiple neurons in the sensorimotor pathway, including several *mod-1*- and *pdfr-1*-expressing neurons, though its role in roaming and dwelling behaviors has not been examined.

Based on AIA's established role in sensory processing and our observations that AIA can drive both roaming- and dwelling-like behaviors, we hypothesized that AIA promotes either roaming or dwelling, depending on the sensory cues in the environment. To test this hypothesis, we examined the foraging behaviors of wild-type animals in different sensory contexts, and compared them to animals in which AIA had been silenced (*AIA::unc-103gf*). Given that AIA responds to food odors, we developed a patch foraging assay in which animals placed on a sparse food patch can navigate a food odor gradient to approach an adjacent dense food patch (*Figure 7A*). This assay is notably different from standard chemotaxis assays, where animals are not in contact with any food source and therefore do not display roaming or dwelling behaviors. We also examined AIA's impact on roaming and dwelling in the absence of an olfactory gradient by performing a second assay where wild-type or AIA-silenced animals were presented with uniform-density bacterial food (*Figure 7F*).

In the patch foraging assay, wild-type animals exhibited directed motion toward the dense food patch and alternated between roaming and dwelling as they approached it (*Figure 7A*, bottom). Compared to control plates without the dense food patch, animals in the patch foraging assay spent more time in the roaming state (*Figure 7B*), and biased their movement towards the dense food patch

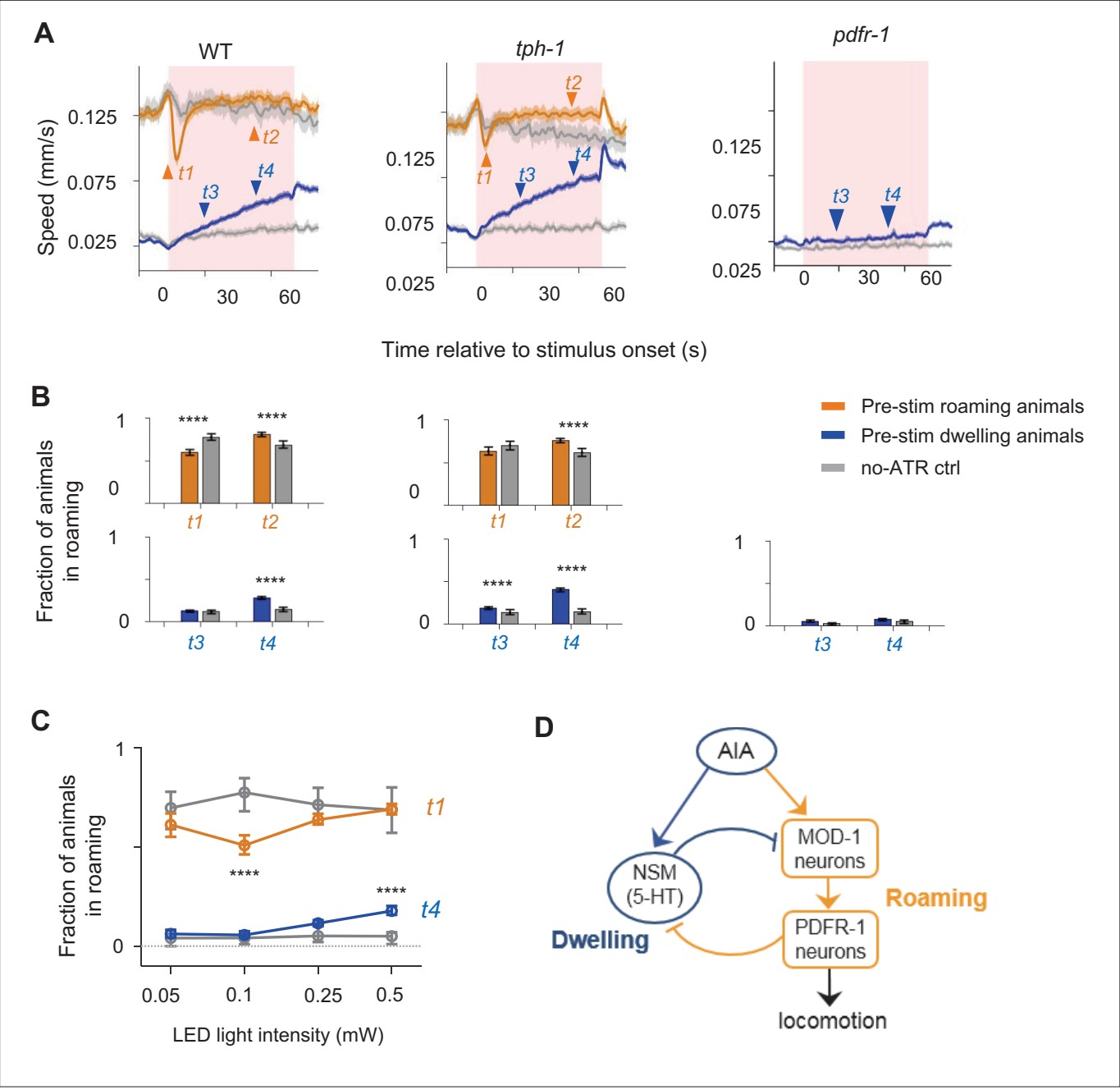

**Figure 6.** The AIA sensory processing neuron can drive behavioral state switching. (**A**) Average locomotion speed before, during, and after AIA::Chrimson activation for wild-type (left), *tph-1* (middle), and *pdfr-1* (right) animals. Animals were grouped by whether they were roaming (orange) or dwelling (blue) prior to AIA stimulation. Pink patches in the background denote the one-minute stimulation window. Gray lines indicate no-all-trans-retinal (no-ATR) controls. N = 1,032 wild-type animals were compared to N = 370 no-ATR controls. N = 927 *tph-1* mutants were compared to N = 284 no-ATR control. N = 383 *pdfr-1* mutants were compared to N = 237 no-ATR controls. Note that roaming in *pdfr-1* animals was too rare and brief to be included for analysis on AIA-induced slowing. Orange and blue arrowheads denote time points used for analyses in (**B**). Error bars are 95% CI of the mean. (**B**) Fraction of animals in the roaming state at different phases of AIA::Chrimson stimulation. Top: Among animals that were roaming pre-stimulation, the fraction of them that were roaming after 4 seconds or 40 seconds from the onset of AIA stimulation. Bottom: Among animals that were dwelling pre-stimulation, the fraction of them that were roaming after 20 seconds or 40 seconds from the onset of AIA stimulation. Same analyses were performed for wild-type (left), *tph-1* (middle), and *pdfr-1* (right) animals. See panel (**A**) for full traces. (**C**) AIA-induced changes in roaming and dwelling at different optogenetic stimulation intensities. For (**B–C**), error bars are 95% CI of the mean and ****P < 0.0001, Wilcoxon rank-sum test. (**D**) Functional architecture of the circuit controlling the roaming and dwelling states, based on results from *Figures 2–5*.

The online version of this article includes the following figure supplement(s) for figure 6:

**Figure supplement 1.** Connectivity of the AIA interneuron.

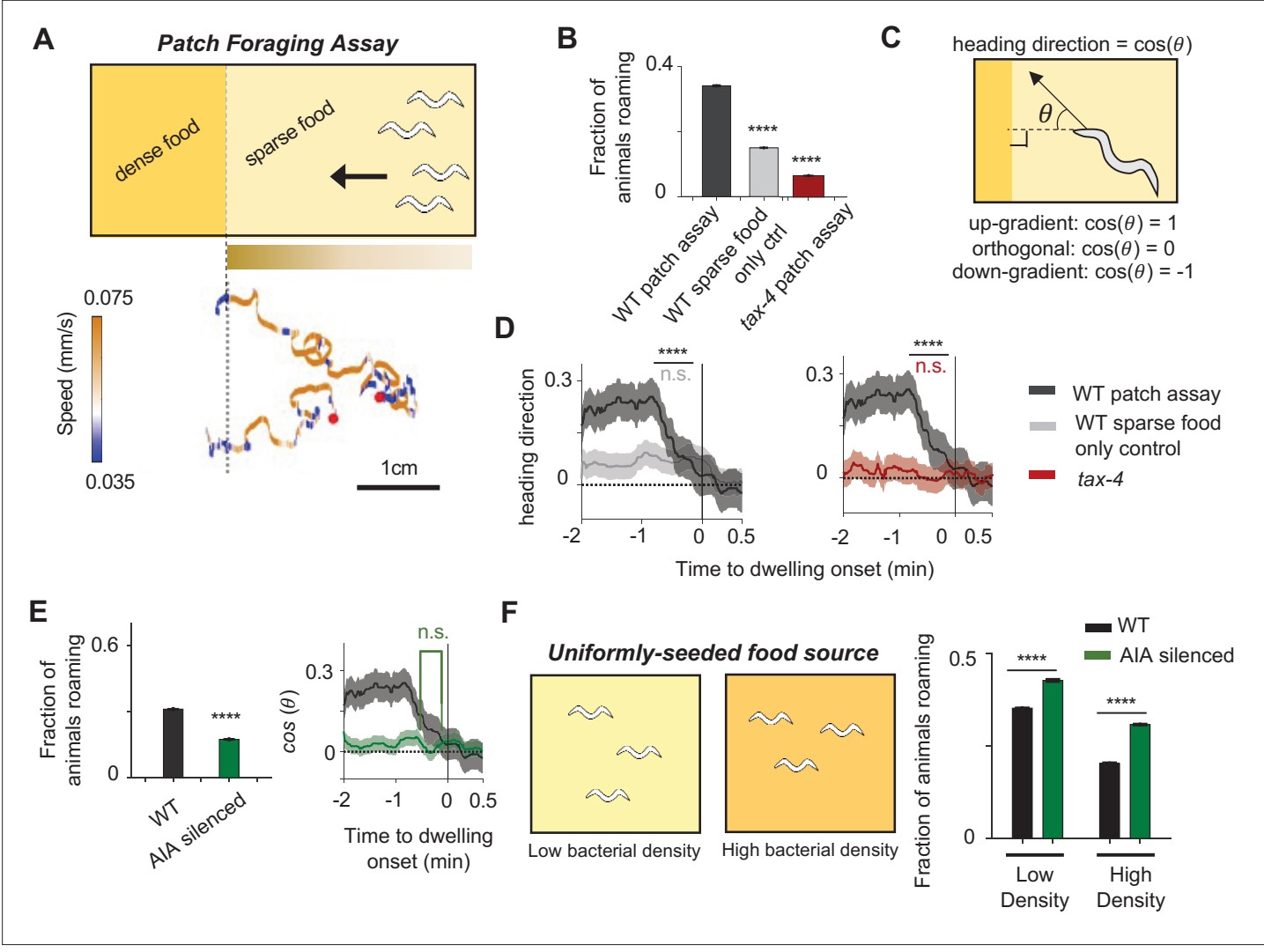

**Figure 7.** The AIA sensory processing neuron can promote either roaming or dwelling, depending on the sensory context. (**A**) Top: Cartoon depicting the patch foraging behavioral assay. Horizontal bar with gradient signifies the food odor gradient emanating from the dense food patch. Bottom: example trajectories of two animals from a patch foraging assay. Color scale indicates speed, with orange corresponding to roaming-like speeds and blue dwelling-like speeds. Red dots denote the starting points of the animals. (**B**) Average fraction of animals roaming on the sparse food patch in the patch foraging assay. Comparisons are made between wild-type animals in the patch foraging assay (n = 288), wild-type animals assayed on uniform sparse food with no dense patch around (n = 194), and *tax-4* animals in the patch foraging assay (n = 81). (**C**) Schematic depicting how heading bias is calculated. (**D**) Event-triggered averages showing average heading bias of animals for two minutes prior to transitions into dwelling states. Experimental conditions are depicted with same color scheme as in (**B**). Data are shown as means ± SEM. The average heading bias within two time windows, one from 60 to 50 s prior to dwelling onset, the other from 20 to 10 s prior to dwelling onset, were compared. (**E**) Left: Average fraction of animals roaming on sparse food in the patch foraging assay in wild-type (black) versus AIA silenced (*AIA::unc-103gf*) animals (green). Right: Heading bias of AIA-silenced animals (green) two minutes prior to the transition into the dwelling state. n = 197. Wild-type data (black with gray error bar) are shown for comparison. (**F**) Left: schematic of behavioral assays in uniformly seeded food environments. Right: Average fractions of animals roaming for wild-type (black bars) and AIA-silenced (green bars) animals exposed to two different densities of uniformly distributed sparse food. For all calculations on fraction of animals roaming, error bars are 95% CI of the mean. For all calculations of heading bias, error bars are SEM. For all comparisons, **p < 0.01, ****p < 0.0001, Wilcoxon rank sum test with BH correction.

The online version of this article includes the following figure supplement(s) for figure 7:

**Figure supplement 1.** Food-directed navigation in patch foraging assays.

as they roamed (*Figure 7—figure supplement 1A*). Animals preferentially switched from roaming to dwelling when their direction of motion (measured as heading bias; *Figure 7C*) began to deviate away from the dense food patch (*Figure 7D*). Because the animal's heading direction impacts the change in odor concentration that it experiences, these results indicate that dynamic changes in the

concentration of food odors influences the transition rates between roaming and dwelling. Consistent this interpretation, we found that chemosensation-defective *tax-4* mutants (*Komatsu et al., 1996*) subjected to the patch foraging assay failed to exhibit elevated roaming and failed to couple the roaming-to-dwelling transition with their direction of motion (*Figure 7B and D*).

We next asked whether AIA was necessary for the sensory-induced modulation of roaming and dwelling states in the patch foraging assay. We found that AIA-silenced animals (*AIA::unc-103gf*) exhibited an overall decrease in roaming compared to wild-type animals and did not selectively enter dwelling states when their movement direction deviated away from the dense food patch (*Figure 7E* and *Figure 7—figure supplement 1B*). These results indicate that AIA is necessary for animals to display elevated roaming in the presence of a food odor gradient and for animals to couple their movement direction with roaming-to-dwelling transitions.

We also examined the roles of 5-HT and PDF in the patch foraging assay. We found that *pdfr-1* mutants failed to increase their roaming in the odor gradient but still displayed some coupling of the roaming-to-dwelling transition to their direction of motion (*Figure 7—figure supplement 1C-D*). In contrast, *tph-1* mutants displayed increased time in the roaming state but did not couple the roaming-to-dwelling transition to their direction of motion (*Figure 7—figure supplement 1C-D*).

Lastly, to examine the role of AIA in controlling roaming and dwelling in the absence of a strong sensory gradient, we compared the behavior of wild-type and AIA-silenced animals in environments with uniformly-seeded bacterial food. We tested two different bacterial densities (*Figure 7F*). In both cases, AIA-silenced animals displayed a significant decrease in the fraction of time spent dwelling (*Figure 7F*). These results suggest that AIA functions to promote the dwelling state in a constant sensory environment. This contrasts sharply with the role of AIA in promoting roaming in the presence of a strong sensory gradient (*Figure 7E*). Taken together, these results indicate that AIA can promote either roaming or dwelling, depending on the overall sensory environment.

## Discussion

Our findings reveal the functional architecture of a neural circuit that generates persistent behavioral states. Circuit-wide calcium imaging during roaming and dwelling identified several neurons whose activities differ between the two states, most notably the serotonergic NSM neuron that displays long bouts of activity during dwelling and inactivity during roaming. By combining circuit imaging with genetic perturbations, we found that mutual inhibition between the serotonergic NSM neuron and the 5-HT and PDF target neurons promotes the stability and mutual exclusivity of the neural activity patterns observed during roaming and dwelling. Furthermore, we found that the AIA sensory processing neuron that responds to food odors sends parallel outputs to both neuromodulatory systems and biases the network towards different states in different sensory contexts. This circuit architecture allows *C. elegans* to exhibit persistent roaming and dwelling states, while flexibly switching between them depending on the sensory context.

### Neural circuit mechanisms that generate persistent behavioral states

The recordings in this study provide new insights into how neural activity changes as animals switch between stable, alternative behavioral states. Previous work had shown that 5-HT and PDF were critical for dwelling and roaming behaviors (*Flavell et al., 2013*), but how they impact circuit activity was not known. We found that NSM displayed long bouts of activity during dwelling and was inactive during roaming. In addition, several neurons that were previously shown to drive forward and reverse movement (*Kato et al., 2015*; *Wang, 2020*; *Roberts et al., 2016*; *Li et al., 2014*; *Luo et al., 2014*; *Chalfie et al., 1985*), including the PDF-producing neuron AVB, were more active during roaming than they were during dwelling. However, whereas NSM displayed long bouts of persistent activity during dwelling, the locomotion neurons displayed fast timescale dynamics associated with forward and reverse movements during roaming, matching their previously described roles in controlling locomotion. *tph-1* mutants that lack 5-HT had an imbalance in the winner-take-all dynamics of this circuit, such that NSM activity was less persistent. *pdfr-1* mutants that lack PDF signaling displayed ectopic co-activation of NSM neurons along with AVB and other roaming-active neurons, as well as exaggerated persistence in NSM. These results suggest that neuromodulation is critical to establish the overall structure of circuit-level activity. Our data also suggest that there is mutual inhibition between NSM

and the neurons that express MOD-1 (an inhibitory 5-HT receptor) and PDFR-1. The MOD-1- and PDFR-1-expressing neurons, which are more active during roaming, synapse onto the PDF-producing neuron AVB that is also more active during roaming, suggesting that they excite AVB. Thus, although NSM and AVB display mutually exclusive high activity states and produce opposing neuromodulators, they do not have direct connections with one another, as is typical in a flip-flop switch. Instead, they coordinate their activities by both interacting with the same network of neurons that expresses the 5-HT and PDF receptors. This architecture might allow for more flexible regulation of behavioral state switching.

The circuit states that correspond to roaming and dwelling differ in several respects. Dwelling states are characterized by persistent activity in serotonergic NSM neurons and reduced activity in several, but not all, locomotion-associated neurons. NSM activation always occurred within seconds of dwelling state onset and persisted for minutes, though NSM inactivation in some cases occurred one or two minutes from roaming state onsets. It is unclear whether this variable time lag involves the perdurance of 5-HT in extracellular space or other effects. Previous work has identified sub-modes of dwelling where animals display distinct subsets of postures (*Cermak et al., 2020*), but our datasets here, which lack detailed posture information, did not permitted us to identify neural correlates of these sub-modes. Roaming states are characterized by fast fluctuations in the activities of neurons that have previously been shown to drive forward (AVB, AIY, RIB) and reverse (AVA) movement (*Kato et al., 2015*; *Wang, 2020*; *Roberts et al., 2016*; *Li et al., 2014*; *Luo et al., 2014*; *Chalfie et al., 1985*). We did not identify a neuron that is persistently active throughout roaming in a manner analogous to NSM activation during dwelling. While it is possible that such a neuron may exist (and that we did not record it in our study), it is also possible that the roaming state might be the 'default' state of the *C. elegans* network and thus does not require devoted, persistently-active neurons to specify the state. Consistent with this possibility, circuit dynamics similar to roaming are observed in the absence of food and even in immobilized animals (*Kato et al., 2015*; *Nguyen et al., 2016*; *Venkatachalam et al., 2016*). The correlational structure of neural activity also differs between roaming and dwelling. For example, the sensory processing neuron AIA is active in both states, but is coupled to NSM during dwelling, and to the forward-active neurons (AVB, AIY, RIB) during roaming. Neurons that can affiliate to different networks and switch their affiliations over time have also been observed in the stomatogastric ganglion and other systems (*Weimann and Marder, 1994*). The correlational changes that we observe here might allow for state-dependent sensory processing.

## Sensory control of roaming and dwelling states

Previous work showed that chemosensory neurons regulate roaming and dwelling behaviors: mutants that are broadly defective in chemosensation display excessive dwelling, while mutants that are defective in olfactory adaptation display excessive roaming (*Fujiwara et al., 2002*). However, the neural circuitry linking sensory neurons to roaming and dwelling had not been characterized. Using a machine learning-based approach, we identified AIA as a pivotal neuron for roaming-dwelling control. AIA receives synaptic inputs from almost all chemosensory neurons in the *C. elegans* connectome and displays robust responses to appetitive food odors (*Larsch et al., 2015*; *Dobosiewicz et al., 2019*). We observed apparently spontaneous AIA dynamics in freely foraging animals, which could reflect responses to small changes in the sensory environment or feedback from other neurons. Here, we found that AIA provides dual outputs to both the dwelling-active NSM neuron and the roaming-active neurons. Three lines of evidence support this interpretation: (1) native AIA activity correlates with NSM during dwelling and with forward-active neurons during roaming, (2) optogenetic activation of AIA can drive behaviors typical of both states, and (3) AIA silencing strongly alters roaming and dwelling states, but has different effects in different sensory contexts: AIA is necessary for roaming while animals navigate up food odor gradients, but is necessary for dwelling while animals are in uniform feeding environments. Thus, AIA is required to couple the sensory environment to roaming and dwelling states.

The dual output of AIA onto both roaming and dwelling circuits is an unusual aspect of the circuit architecture uncovered here. However, similar functional architectures, where a common input drives competing circuit modules, have been suggested to underlie behavior selection in other nervous systems (*Anderson, 2016*; *Seo et al., 2019*; *Jovanic et al., 2016*). One possible function of this motif in the roaming-dwelling circuit is that it might allow both the roaming- and dwelling-active neurons

to be latently activated when the animal is exposed to food odors detected by AIA. AIA-transmitted information about food odors could then be contextualized by other sensory cues that feed into this circuit. For example, NSM is not directly activated by food odors, but instead is directly activated by the ingestion of bacteria via its sensory dendrite in the alimentary canal (*Rhoades et al., 2019*). Thus, when animals detect an increase in food odors that is accompanied by increased ingestion, this might promote dual AIA and NSM activation to drive robust dwelling states. In contrast, when animals detect an increase in food odors that is not accompanied by increased ingestion, this might activate AIA and the other side of the mutual inhibitory loop, biasing the animal towards roaming. This flexible architecture could therefore allow animals to make adaptive foraging decisions that reflect their integrated detection of food odors, food ingestion, and other salient sensory cues.

### Mutual inhibition as a network motif for generating opposing activity states

Long-standing theoretical work (*Major and Tank, 2004*; *Wang, 2012*; *Seung, 1996*; *Xiong and Ferrell, 2003*) and recent experimental evidence (*Wang, 2012*; *Goldman et al., 2007*) has highlighted the role of recurrent circuitry in driving persistent neural activity. In particular, mutual inhibition has long been proposed to underlie opposing cognitive states (*Machens et al., 2005*; *Xiong and Ferrell, 2003*; *Goldman et al., 2007*; *Wang, 2001*). Recent modeling and experimental studies of the locomotion circuit of *C. elegans* has shown that fast timescale behavioral changes involve stochastic switching of flip-flop circuits, and nested oscillatory dynamics that depend on the ongoing state of these circuits (*Wang, 2020*; *Roberts et al., 2016*; *Kaplan et al., 2020*). Our results here suggest that a neural circuit with mutual inhibition mediated by neuromodulatory signals can generate behavioral switching over a much longer timescale, giving rise to persistent behavioral states that can be flexibly generated depending on the demands of the sensory environment.

## Materials and methods

### Key resources table

| Reagent type (species) or resource | Designation | Source or reference | Identifiers | Additional information |
| --- | --- | --- | --- | --- |
| Strain, strain background (*C. elegans*) | N2 | CGC | ID_FlavellDatabase: N2 | Wild-type Bristol N2 |
| Strain, strain background (*C. elegans*) | SWF90 | This study | ID_FlavellDatabase: SWF90 | *flvEx46[[tph-1::GCaMP6m, mod-1::GCaMP6m, sto-3::GCaMP6m, glr-3::GCaMP6m, odr-2b::GCaMP6m, gcy-28.d::GCaMP6m, lgc-55(short)::GCaMP6m, nmr-1::GCaMP6m, tph-1::wrmScarlett, mod-1::wrmScarlett, nmr-1::wrmScarlett, sto-3::wrmScarlett]; lite-1(ce314), gur-3(ok2245)]*. See: **Figure 1**. |
| Strain, strain background (*C. elegans*) | SWF113 | This study | ID_FlavellDatabase: SWF113 | *flvIs1[tph-1::GCaMP6m, mod-1::GCaMP6m, sto-3::GCaMP6m, glr-3::GCaMP6m, odr-2b::GCaMP6m, gcy-28.d::GCaMP6m, lgc-55(short)::GCaMP6m, nmr-1::GCaMP6m, tph-1::wrmScarlett, mod-1::wrmScarlett, nmr-1::wrmScarlett, sto-3::wrmScarlett]; lite-1(ce314), gur-3(ok2245)*. See: **Figures 1–5**, **Figure 1—figure supplements 1–8**, **Figure 2—figure supplement 1**, **Figure 3—figure supplement 1**, **Figure 4—figure supplements 2–5**, **Figure 5—figure supplements 1–3**. |
| Strain, strain background (*C. elegans*) | SWF186 | This study | ID_FlavellDatabase: SWF186 | *flvIs1; lite-1(ce314); gur-3 (ok2245); mod-1(ok103)*. See: **Figures 3–4**, **Figure 3—figure supplements 1–2**, **Figure 4—figure supplements 4–5**. |
| Strain, strain background (*C. elegans*) | SWF124 | This study | ID_FlavellDatabase: SWF124 | *flvIs1; lite-1(ce314); gur-3 (ok2245); pdfr-1(ok3425)* See: **Figure 4**, **Figure 4—figure supplements 1–5**. |
| Strain, strain background (*C. elegans*) | SWF263 | This study | ID_FlavellDatabase: SWF263 | *flvIs1; lite-1(ce314); gur-3 (ok2245); flvEx129[pdfr-1::acy-1gf, elt-2::nGFP]*. See: **Figure 4**, **Figure 4—figure supplements 1–5**. |

*Continued on next page*

*Continued*

| Reagent type (species) or resource | Designation | Source or reference | Identifiers | Additional information |
|---|---|---|---|---|
| Strain, strain background (*C. elegans*) | SWF125 | This study | ID_FlavellDatabase: SWF125 | *flvEx46; lite-1(ce314); gur-3 (ok2245); tph1(mg280); pdfr-1(ok3425).* See: **Figures 3–4**, **Figure 3—figure supplements 1–2**, **Figure 4—figure supplements 2–5** |
| Strain, strain background (*C. elegans*) | SWF168 | This study | ID_FlavellDatabase: SWF168 | *flvIs1; lite-1(ce314); gur-3 (ok2245); flvEx86[tph-1(short)::chrimson, elt-2::nGFP].* See: **Figure 2**, **Figure 2—figure supplement 1**, **Figure 4—figure supplement 5**. |
| Strain, strain background (*C. elegans*) | SWF801 | This study | ID_FlavellDatabase: SWF801 | *flvIs1; lite-1(ce314); gur-3(ok2245); mod-1(ok103); flvEx86[tph-1(short)::chrimson, elt-2::nGFP].* See: **Figure 2—figure supplement 1**. |
| Strain, strain background (*C. elegans*) | CX14684 | This study | ID_FlavellDatabase: CX14684 | *pdfr-1(ok3425); kyIs580[mod-1::nCre, myo-2::mCherry]; kyEx4816[ttx-3::ChR2(C128S)-GFP, odr-2b::inv[ChR2-sl2-GFP], myo-3::mCherry].* See: **Figure 4**. |
| Strain, strain background (*C. elegans*) | SWF194 | This study | ID_FlavellDatabase: SWF194 | *flvIs1; lite-1(ce314); gur-3 (ok2245); flvEx98[gcy-28.d::Chrimson, elt-2::nGFP].* See **Figure 6**. |
| Strain, strain background (*C. elegans*) | SWF216 | This study | ID_FlavellDatabase: SWF216 | *flvIs1; lite-1(ce314); gur-3 (ok2245);tph-1(mg280); flvEx98[gcy-28.d::Chrimson, elt-2::nGFP].* See **Figure 6**. |
| Strain, strain background (*C. elegans*) | SWF326 | This study | ID_FlavellDatabase: SWF326 | *flvIs1; lite-1(ce314); gur-3 (ok2245); pdfr-1(ok3425); flvEx98[gcy-28.d::Chrimson, elt-2::nGFP].* See **Figure 6**. |
| Strain, strain background (*C. elegans*) | CX14597 | **Larsch et al., 2015** | ID_FlavellDatabase: CX14597 | *kyEx4745[gcy-28.d::unc-103gf::sl2-mCherry, elt-2::mCherry].* See **Figure 7**, **Figure 7—figure supplement 1**. |
| Strain, strain background (*C. elegans*) | CX13078 | This study | ID_FlavellDatabase: CX13078 | *tax-4(p678) [5 x backcrossed to N2].* See **Figure 7**, **Figure 7—figure supplement 1**. |
| Strain, strain background (*C. elegans*) | CX14295 | **Flavell et al., 2013** | ID_FlavellDatabase: CX14295 | *pdfr-1(ok3425).* See: **Figure 7—figure supplement 1**. |
| Strain, strain background (*C. elegans*) | MT15434 | CGC | ID_FlavellDatabase: MT15434 | *tph-1(mg280).* See: **Figure 7—figure supplement 1**. |
| Strain, strain background (*C. elegans*) | SWF392 | This study | ID_FlavellDatabase: SWF392 | *lite-1(ce314); gur-3(ok2245); flvEx148[gcy-28.d::Chrimson, myo-3::mCherry].* See **Figure 6**. |
| Strain, strain background (*C. elegans*) | SWF167 | This study | ID_FlavellDatabase: SWF167 | *flvIs1; lite-1(ce314), gur-3(ok2245); flvEx85[pdf-1::Chrimson, elt-2::nGFP].* See: **Figure 4**, **Figure 4—figure supplement 5**. |
| Strain, strain background (*C. elegans*) | SWF166 | This study | ID_FlavellDatabase: SWF166 | *flvIs1; lite-1(ce314), gur-3(ok2245); flvEx84[mod-1::Chrimson, elt-2::nGFP].* See: **Figure 3**, **Figure 4—figure supplement 5**. |
| Software, algorithm | MATLAB | MathWorks (https://www.mathworks.com) | RRID:SCR_001622 | v2019a |
| Software, algorithm | NIS Elements | Nikon (https://www.nikoninstruments.com/products/software) | RRID:SCR_014329 | v5.02.00 |
| Software, algorithm | Streampix | Norpix (https://www.norpix.com/products/streampix/streampix.php) | RRID:SCR_015773 | v7.0 |

## Growth conditions and handling

Nematode culture was conducted using standard methods (*Brenner, 1974*). Populations were maintained on NGM agar plates with *E. coli* OP50 bacteria. Wild-type was *C. elegans* Bristol strain N2. For genetic crosses, all genotypes were confirmed using PCR. Transgenic animals were generated by injecting DNA clones plus fluorescent co-injection marker into gonads of young adult hermaphrodites. One day old hermaphrodites were used for all assays. All assays were conducted at room temperature (~22 °C).

## Construction and characterization of multi-neuron GCaMP strain

To generate a transgenic strain expressing GCaMP6m in a specific subset of neurons involved in roaming and dwelling, we first generated pilot strains where one or two plasmids were injected at a time to optimized DNA concentrations. This also allowed us to determine the precise GCaMP6m and/or Scarlett expression pattern from each promoter. We then injected these plasmids as a mixture into *lite-1;gur-3* double mutants, which are resistant to blue light delivered during calcium imaging. We selected a line for use that had normal behavioral parameters and showed relatively balanced expression of GCaMP6m and Scarlett in the target cells (SWF90). To obtain more consistent expression, the transgene was integrated by UV to generate *flvIs1* (SWF113). The integrated strain was outcrossed four times.

## Microscope design and assembly

### Overview

The tracking microscope design was inspired and based off previously described systems (*Faumont and Lockery, 2006*; *Nguyen et al., 2016*; *Venkatachalam et al., 2016*), with several modifications aimed at reducing motion artifacts and extending the duration of calcium imaging, so that long-lasting behavioral states could be examined. As illustrated in *Figure 1—figure supplement 1*, two separate light paths, below and above the specimen, were built onto a Ti-E inverted microscope (Nikon).

### High-magnification light path for GCaMP imaging

The light path used to image GCaMP6m and Scarlett at single cell resolution is an Andor spinning disk confocal system. Light supplied from a 150 mW 488 nm laser and a 50 mW 560 nm laser passes through a 5000 rpm Yokogawa CSU-X1 spinning disk unit with a Borealis upgrade (with a dual-camera configuration). A 40 x/1.15NA CFI Apo LWD Lambda water immersion objective (Nikon) with a P-726 PIFOC objective piezo (PI) was used to image the volume of the worm's head. A custom quad dichroic mirror directed light emitted from the specimen to two separate Andor Zyla 4.2 USB3 cameras, which had in-line emission filters (525/50, and 625/90). Data was collected at 2 × 2 binning in a 512 × 512 region of interest in the center of the field of view.

### Low-magnification light path for closed-loop tracking

A second light path positioned above the animal collected data for closed-loop tracking. Light supplied from a Sola SE2 365 Light Engine (Lumencor) passed through a DSRed (49005, Chroma) filter set and a 10 x/0.3NA air objective to excite Scarlett in the head of the worm. Red light emitted from the specimen passed through the filter set to an acA2000-340km Basler CMOS camera. Data was collected at 100 Hz.

### Synchronized control of camera exposures and illumination light sources

The Andor Zyla cameras used for calcium imaging were run in rolling shutter mode. A trigger signal was generated by one of the two cameras whenever the camera shutter is fully open (~2 ms per exposure). This trigger signal served as a master control that synchronized several devices (*Figure 1—figure supplement 1B*). First, it was used to drive the 488 nm and 560 nm lasers, such that illumination is only provided when the full field of view is open. Second, the same trigger signal was used controlled the movement of the objective piezo, such that fast piezo movement occurs largely outside the window of laser illumination. Lastly, this signal was used to time the green LED used by the closed-loop tracking system. The LED was turned on only when the calcium imaging cameras were not actively acquiring images (i.e. outside the window when the rolling shutter is fully open) and when the lasers were off.

Together, these approaches minimize photo-bleaching, photo-toxicity, and motion artifacts induced by movable parts of the microscope.

## Closed-loop tracking software

A custom C/C ++ software was used to process incoming frames from the tracking camera and to instruct the movement of a motorized stage (96S107-N3-LE2, Ludl; with a MAC6000 controller) to keep the head region of the animal at the center in the field of view. This software was adapted from Nguyen et al. with two key modifications: First, at each control cycle, the future velocity of the stage was calculated to match the predicted future velocity of the animal (i.e. predictive control as opposed to proportional control employed in previous study). Specifically,

$$v_{stage}\left(t+1\right) = v_{animal}\left(t\right) + \frac{\left(v_{animal}\left(t\right) - v_{animal}\left(t-1\right)\right)}{t}$$

where $v_{stage}\left(t\right)$ is the instantaneous velocity of the stage and $v_{animal}\left(t\right)$ the instantaneous velocity of the animal. The latter was estimated as described below (see **Estimation of instantaneous animal location and velocity**). The right side of the formula was found empirically to be sufficient for predicting future animal velocity. The second modification was that we used the motion of the head region of the animal to extrapolate the locomotory state of the animal. This approach results in a loss of postural information, but circumvents the need for a third light path for imaging the full body of the animal.

## Optogenetic stimulation during calcium imaging

For experiments where we activated Chrimson while performing circuit-wide imaging, L4 animals were picked to plates with 1 μM ATR (or not, in the case of -ATR controls) for overnight growth. They were then subjected to our standard calcium imaging methods described above, except the 561 nm laser light to image Scarlett in neurons was omitted due to concerns of cross-activation of Chrimson. For optogenetic stimulation, an overhead spotlight LED (Mightex 617 nm Type-H) directed toward the sample was illuminated for 30 s at a time. Animals were stimulated one to three times each.

## Behavioral assays

### Patch foraging assay

For the patch foraging assays, we used 24.5cm x 24.5 cm NGM plates. Plates were uniformly seeded with sparse OP50 bacteria (OD 0.5 diluted 300 x), and one half of the plate was seeded with dense bacteria (OD 0.5 concentrated 20 x). The border between the sparse and dense food was always sharp and typically very straight. Plates were left overnight at room temperature. The following day, 1-day-old adult animals were picked to the sparse side of the food plate, approximately 1.5 cm from the dense food patch. Video recordings were started immediately, although for all analyses the first 20 min of data (equilibration time) was not analyzed. Videos were recorded at three fps using Streampix 7.0, a JAI SP-20000M-USB3 CMOS cameras (41 mm, 5120 × 3840, Mono) with a Nikon Micro-NIKKOR 55 mm f/2.8 lens. Backlighting was achieved using a white panel LED (Metaphase Technologies Inc White Metastandard 10" X 25", 24VDC). Assay plates were placed on glass 3" above LEDs to avoid heat transfer to plates. Videos were processed using custom Matlab scripts (all code is available at https://github.com/neurourchin/Flexible-ctrl-persistent-states_Ji-et-al-eLife-2021 ; *Ji, 2021*, copy archived at swh:1:rev:a459ec849a1e6dd2a409691772ec23a660b14edd), which included a step to manually confirm the exact frame of dense food patch encounter for each animal. Segmentation of behavior into roaming and dwelling was conducted as previously described (*Flavell et al., 2013*).

### Foraging at different food densities

To examine animal behavior in uniform environments with different food densities, we seeded NGM plates (either circular 10 cm or 24.5 × 24.5 cm) with different densities of food. For the experiments in *Figure 7*, low-density was OP50 bacteria at OD 0.5 diluted 300 x; high-density was OD 0.5 concentrated 20 X. Plates grew overnight at room temperature. The following day, 1-day-old adult animals were picked to these plates and allowed to equilibrate for 45 min, after which video recordings began. Videos were recorded and analyzed as described above.

## Optogenetic stimulation during foraging behavior

For optogenetic stimulation of free-behaving animals, we picked one-day-old adult animals (grown on 50 uM ATR the night before) to NGM plates seeded with 300 X diluted OD 0.5. Animals were permitted to equilibrate for 45 min, after which videos were recorded using the setup described above. In these videos, light for optogenetic stimulation was delivered using a 625 nm Mightex BioLED at 30 uW/mm$^2$. Patterned light illumination was achieved using custom Matlab scripts, which were coupled to a DAQ board (USB-6001, National Instruments) and BioLED Light Source Control Module (Mightex). Videos were analyzed as described above.

# Data analysis for calcium imaging

## Semi-automated image segmentation to obtain neuron outlines

All image analyses were performed on maximum intensity projections of the collected z-stacks, since the neurons were well separated along the x-y axes. We used data from the side of the animal (left or right) that was closest to the microscope objective, due to better optical quality of these data. First, feature points and feature point descriptors were extracted for each frame of the calcium imaging video. Next, an N-by-N similarity matrix (N = number of frames in a video) was generated where each entry equals the number of matched feature points between a pair of frames. The columns of this matrix were clustered using hierarchical clustering. Around 30 frames (typically 1–2% of frames from a video) were chosen across the largest 15 clusters. These frames were then segmented manually. The user was asked to outline the region for interest (ROI) around each neuronal structure of interest (axonal segment for the AIY neurons; soma for all other neurons). After manual segmentation, the automatic segmentation software looped through each of the remaining frames. For each unsegmented frame (target frame), a best match (reference frame) was found among the segmented frames based on the similarity matrix. Then, geometrical transformation matrices were estimated using the locations of the matched feature points. The estimated transformation was then applied to the boundary vertices of each ROI in the reference frame to yield the estimated boundary of the same region in the target frame. Once done, the target frames with their automatically computed ROIs were included into the pool of segmented frames and could serve as a reference frame for the remaining unsegmented frames. This procedure was repeated iteratively through the rest of the video.

## Calcium signal extraction and pre-processing

The fluorescent signal from each neuron in a given frame was calculated as the median of the brightest 100 pixels within the ROI (or all pixels if the size of the ROI was smaller than 100 pixels) of that neuron. This approach was adopted to render the estimation of calcium signal insensitive to the exact segmentation boundary of the neuron, which could inadvertently contain background pixels. This was done for both the green and the red channels. The following pre-processing steps were then applied to the time-series signals from both channels: (1) To reduce spurious noise, a sliding median filter with a window size of 5 frames were applied to the time series (*Figure 1—figure supplement 2D*). (2) To correct for the decay in fluorescent signal due to photobleaching, an exponential function was first fit to the time series. Next, the estimated exponential was normalized by its initial value and divided away from the denoised time series (*Figure 1—figure supplement 2E*). (3) To control for fluctuations in fluorescent signal due to the movement of the animal, we calculated the ratiometric signal. Specifically, the denoised and bleach-corrected time series from the green channel was divided by that from the red channel. (4) Lastly, to control for the variations in the dynamic range of the calcium signal due to variations in the expression of the fluorescent indicators, we normalized the ratio-metric signal such that the 1st percentile of the signal takes a value of 0 while the 99th percentile takes the value of 1. To control for cases where a given neuron never became active in a given recording (e.g. NSM in *pdfr-1::acy-1gf* animals), exceptions were made if a neuron's peak activity in a given recording was less than 10 % of the average across all recordings. In this case, the original $\Delta R/R_0$ value was used without normalization. Apart from this exception, the normalized ratio-metric signal was used for subsequent data analyses, except where indicated. These data processing steps (dividing by mScarlett; normalizing to a 0–1 scale) did not change the distributions of GCaMP intensity values (*Figure 1—figure supplement 2H*).

## Estimation of instantaneous animal location and velocity

The instantaneous location of the animal $\begin{bmatrix} x_a \\ y_a \end{bmatrix}$ was calculated based on the following formula:

$$\begin{bmatrix} x_a \\ y_a \end{bmatrix} = \begin{bmatrix} x_s \\ y_s \end{bmatrix} + r * \begin{bmatrix} cos\theta & -sin\theta \\ sin\theta & cos\theta \end{bmatrix} \cdot \begin{bmatrix} x_c \\ y_c \end{bmatrix}$$

where $\begin{bmatrix} x_s \\ y_s \end{bmatrix}$ is the instantaneous location of the microscope stage, $\begin{bmatrix} x_c \\ y_c \end{bmatrix}$ is the position of the head region of the animal as seen on the frame captured by the tracking camera, $\theta$ is the rotation angle between the field of view of the tracking microscope and the sensor of the tracking camera, and $r$ is the pixel size of the frames taken through the tracking camera. The velocity of the animal was calculated by dividing the displacement vector of the animal between adjacent time points by the duration of the time interval.

## Classification of roaming and dwelling states

Previous methods to segment roaming and dwelling defined these states based on the speed and angular speed of animal movement measured over 10 s time windows (**Ben Arous et al., 2009**; **Flavell et al., 2013**). These prior datasets were recorded on multi-worm trackers with lower resolution than that of our confocal microscope. High-amplitude angular speed measurements from the low-resolution trackers primarily reflected paused movement and/or low-speed forward/backward movement. Under high-resolution confocal recordings, angular speed was measured with greater precision and, thus, displayed a different profile, in part reflecting body oscillations. Therefore, we utilized a slightly different approach to segment roaming and dwelling from our confocal recordings: (i) we computed the median and variance of animal speed using a sliding window of 20 s, which transformed the 1-dimensional speed data into two dimensions (**Figure 1—figure supplement 4A**). We then (ii) fit a Hidden Markov Model with Gaussian mixture emissions to this two-dimensional dataset. This yielded a model with three Gaussian components and two hidden states. The two hidden states successfully captured periods of persistent fast and slow movements, which we define as roaming and dwelling, respectively (**Figure 1** and **Figure 1—figure supplement 3**). Roaming and dwelling segmentation of multi-worm tracking data (**Figures 6–7**) was performed using previous methods (**Flavell et al., 2013**) and is described above.

## Aligning calcium imaging data with behavioral data

As described in the Microscope Design and Assembly section, the trigger signal for the confocal laser was simultaneously sent to the computer controlling the tracking microscope. This computer thereby stores two sets of time stamps, one for the laser illumination sequence and the other for the behavioral tracking video. Since the internal clock is the same, we can interpolate both the calcium activity data and the behavioral data onto the same time axis. Specifically, we interpolated both the calcium activity and behavior time series to obtain a common sampling frequency of 2 Hz.

## Principal component analysis (PCA)

An *N-by-M* data matrix was assembled with the rows representing neuron identity (N = 10) and the columns time points. Data across different recording sessions were concatenated together along the time dimension. PCA was performed by first subtracting the mean from each row and then applying singular value decomposition to the matrix. We chose this method over the previously described approach of performing PCA on the time derivatives of the calcium signals (**Kato et al., 2015**). We found that applying PCA on the time derivatives did not yield PCs with intuitive behavioral correlates when applied to our data. This may have resulted from the higher sensitivity of the time derivatives to measurement noise in these freely moving animals or from our recordings consisting of a different subset of neurons, compared to previous studies.

## Cross-correlation in neural activity

To estimate the time-lagged similarity between the activity of two neurons for a given genotype, the cross-correlation function (XCF) was first calculated individually for each data set of that genotype and then averaged. Bootstrapping was done to obtain confidence intervals on the mean. To examine the functional coupling between two neurons over time, average XCFs were calculated for data from a series of 60 s time windows spanning from 90 s before NSM activation to 90 s after. For each time window, the point with the largest absolute value along the average XCF was identified. The mean and 95% CI values of these extrema points were concatenated chronologically to generate plots.

## Classification of NSM activity states

We first computed the local median and variance of NSM activity using a sliding window of 20 s. This transformed the one-dimensional activity data into two dimensions (*Figure 1—figure supplement 4B*). We then fit a Hidden Markov Model with Gaussian mixture emissions to this 2-dimensional dataset. This yielded a model with 3 Gaussian components and two hidden states. As the average NSM activity under these two hidden states differed significantly, we interpreted these hidden states as states of high and low NSM activity.

## General linear model to predict animal speed from calcium activity

The model performs linear regression on a set of linear and nonlinear terms derived from the instantaneous calcium activity of individual neurons. These include an intercept term, linear and squared terms of each neuron's activity, and all pairwise products of neural activity across all 10 neurons measured. The model then computes a linear fit of these predictor variables to the concurrent speed of the animal using QR decomposition.

## Logistic regression to predict foraging state from calcium activity

Logistic regression was performed using the instantaneous activity of all or a subset of the 10 neurons to predict the concurrent foraging state (i.e. roaming or dwelling). Model parameters was regularized through the elastic net algorithm, which implements a combination of $L^1$ and $L^2$ normalization.

## Segmentation of NSM and AVB activities via thresholding

For analyses in *Figure 4*, we segmented NSM and AVB activities into high versus low values. The threshold values for defining the high versus low values were determined using the Otsu method. This was implemented using the "multithresh" function in MATLAB (with the source data set to the wild type activity of NSM or AVB, and the parameter N set to 1). Thresholds determined from wild-type animals were uniformly applied to all genotypes.

## Convolutional neural network (CNN) classifier

The classifier was implemented using the Deep Learning Toolbox in MATLAB. The architecture of the network consists of a single convolutional layer with a single channel of two 9-by-3 convolutional kernels with no padding, followed by a Rectified Linear Unit (ReLu) layer, a fully connected layer with two neurons, a two-way softmax layer and a classification output layer. The last layer is specifically required for the Matlab implementation and computes the cross-entropy loss. We used two 9-by-3 convolutional kernels to allow for the possibility that two separate activity patterns might be necessary for accurate predictions, though in reality only one convolutional kernel had informative values (the other was typically comprised of values close to zero). Calcium activity from all neurons imaged, except for the 5-HT neuron NSM, were used for training, validation and testing. To specifically predict the transition from roaming to dwelling, only data during roaming were used to predict the onset of NSM activity. For each wild-type data set, calcium activity during each roaming state was first downsampled by averaging data from time bins of various widths (7.5–50 s, see *Figure 5—figure supplement 1*) starting from immediately prior to the onset of a dwelling state and going back in time to the beginning of the roaming state. The 30 s bin width was selected after a systematic examination of how well CNNs performed when trying a range of different bin widths and total numbers of bins (*Figure 5—figure supplement 1*). Each data point in the down-sampled data was assigned a label of 1 or 0: 1 if it is immediately prior to an episode of NSM activation, and 0 otherwise. Positive and

negative samples were balanced by weighting the prediction error of each sample by the number of samples in that class. The positive and negative sample groups were each partitioned at random into the training, validation, and test sets at an 8:1:1 relative ratio. This random partition was repeated 200 times. For each data partition, network training was performed 10 times with random initial conditions, using Stochastic Gradient Descent with Moment (SGDM) with the following hyper-parameters:

| Hyper-parameter name | Value |
|---|---|
| Initial Learning Rate | 0.09 |
| L2 Regularization Rate | 0.0001 |
| Learning Rate Drop Factor | 0.1 |
| Learning Rate Drop Period | 10 |
| Momentum | 0.9 |
| Validation Frequency | 30 |
| Max number of epochs | 150 |

To identify convolutional kernels that consistently contribute to classifier accuracy, convolutional kernels from networks that achieved greater than 50 % test accuracy were recorded and k-means clustering was performed. Within each cluster, the distribution of weights at each kernel location was used to extract a confidence interval for the mean value of that kernel element. Elements of the kernel with mean values significantly different from 0 were taken to indicate important neural activity profiles for predicting NSM activation. Since each kernel element maps to the activity of a given neuron at a particular time window, the preferred sign of a kernel element would suggest whether a neuron is preferentially active (when the preferred sign is positive) or inactive (when the preferred sign is negative) at that time window.

Feature selection was performed to identify key neurons whose activity critically contribute to classification accuracy. To generate the results in *Figure 5B*, data from a chosen neuron was removed from the 9-neuron data set, and the resulting partial data set was used to train CNNs following the procedure described above. To generate the results in *Figure 5—figure supplement 2B*, two types of partial data sets were used. In the first category, data from six out of nine neurons were used for training. We tested all possible 9-choose-6 neuron combinations. In second category, we tested using data from only RIB, AIA, and AVA for network training.

## Data analysis for behavioral assays

### Extraction of locomotory parameters
Animal trajectories were first extracted using custom software described previously (*Rhoades et al., 2019*). Speed and angular speed were calculated for all time points of each trajectory, and then averaged over 10 second intervals.

### Identification of roaming and dwelling states
Roaming and dwelling states were identified as previously described (*Flavell et al., 2013*). Briefly, the speed and angular speed measured for each animal at each time point was assigned into one of two clusters. This allowed each animal trajectory to be converted into a binary sequence. A two-state HMM was fit to these binary sequences to estimate the transition and emission probabilities. This was done separately for each genotype under each experimental condition.

### Calculation of heading bias

The instantaneous heading bias $c(t)$ was defined as:

$$c(t) = \frac{(v \cdot g)}{(\|v\| \times \|g\|)}$$

where $v$ is the instantaneous velocity of the animal, and $g$ is the unit vector that points from the animal's current location to the nearest point on the boundary between the sparse food patch and the

dense food patch. Here, $g$ is used as the proxy for the gradient of olfactory cues at the animal's current location. Equivalently, $c(t)$ is the cosine of the angle between the animal's instantaneous direction of motion and the direction of the chemotactic gradient at its current location.

## Statistical analysis

### Comparison of sample means

The Wilcoxon ranksum test was applied pair-wise to obtain the raw p-values. When multiple comparisons were done for the same type of experiment (e.g. comparing the fraction of animal roaming during the patch foraging assay for different genotypes), the Benjamini-Hochberg correction was used to control the false discovery rate. A corrected p-value less than 0.05 was considered significant.

### Bootstrap confidence intervals

Bootstrapping was performed by sampling with replacement $N$ times from the original data distribution ($N$ equals the size of the original distribution). This procedure was repeated 1000 times and the test statistic of interest (e.g. the sample mean) was calculated each time on the bootstrapped data. The 5th and 95th percentiles of the calculated values then constitute the lower and upper bounds of the 95 % confidence interval.

## Acknowledgements

We thank Rachel Wilson, Andrew Gordus, Paul Greer, Yun Zhang, Michael Hendricks, Mike O'Donnell, Dipon Ghosh, and members of the Flavell lab for helpful comments on the manuscript. We thank Andrew Leifer for helpful advice on and sharing software related to the tracking microscope, Thomas Boulin for sharing the mScarlett plasmid, and Nate Cermak for help with hardware control on the tracking microscope. We thank the Bargmann lab and the *Caenorhabditis* Genetics Center (supported by P40 OD010440) for strains. NJ acknowledges support from the Picower Fellows program and the Charles King Trust Postdoctoral Fellowship. SWF acknowledges funding from the JPB Foundation, PIIF, PNDRF, the NARSAD Young Investigator Award Program, McKnight Foundation, the Alfred P Sloan Foundation, NIH (R01NS104892) and NSF (IOS 1845663 and DUE 1734870).

## Additional information

### Funding

| Funder | Grant reference number | Author |
| --- | --- | --- |
| National Institute of Neurological Disorders and Stroke | R01NS104892 | Steven W Flavell |
| National Science Foundation | IOS 1845663 | Steven W Flavell |
| National Science Foundation | DUE 1734870 | Steven W Flavell |
| JPB Foundation | PIIF | Steven W Flavell |
| JPB Foundation | PNDRF | Steven W Flavell |
| Brain and Behavior Research Foundation | NARSAD Young Investigator Award | Steven W Flavell |
| McKnight Foundation | McKnight Scholars Award | Steven W Flavell |
| JPB Foundation | Picower Fellowship | Ni Ji |
| Alfred P. Sloan Foundation | Sloan Research Fellowship | Steven W Flavell |
| Charles A. King Trust | Charles King Trust Postdoctoral Fellowship | Ni Ji |

| Funder | Grant reference number | Author |
|--------|------------------------|--------|

The funders had no role in study design, data collection and interpretation, or the decision to submit the work for publication.

## Author contributions

Ni Ji, Conceptualization, Data curation, Formal analysis, Investigation, Methodology, Project administration, Software, Supervision, Validation, Visualization, Writing - original draft, Writing - review and editing; Gurrein K Madan, Conceptualization, Investigation, Methodology, Writing - review and editing; Guadalupe I Fabre, Software; Alyssa Dayan, Formal analysis, Investigation, Software; Casey M Baker, Formal analysis, Investigation, Writing - review and editing; Talya S Kramer, Ijeoma Nwabudike, Investigation, Writing - review and editing; Steven W Flavell, Conceptualization, Data curation, Formal analysis, Funding acquisition, Investigation, Methodology, Project administration, Supervision, Visualization, Writing - original draft, Writing - review and editing

## Author ORCIDs

Ni Ji (ID) http://orcid.org/0000-0002-7870-0678
Steven W Flavell (ID) http://orcid.org/0000-0001-9464-1877

## Decision letter and Author response

Decision letter https://doi.org/10.7554/eLife.62889.sa1
Author response https://doi.org/10.7554/eLife.62889.sa2

# Additional files

## Supplementary files

• Transparent reporting form

## Data availability

Code has been made available on Github at https://github.com/neurourchin/Flexible-ctrl-persistent-states_Ji-et-al-eLife-2021 copy archived at https://archive.softwareheritage.org/swh:1:dir:144358b9cc82a172101192d73eb318c68f5c6467. Data has been made available on Dryad.

The following dataset was generated:

| Author(s) | Year | Dataset title | Dataset URL | Database and Identifier |
|-----------|------|---------------|-------------|-------------------------|
| Ji N, Madan G, Fabre G, Dayan A, Baker C, Nwabudike I, Flavell S | 2020 | A neural circuit for flexible control of persistent behavioral states | https://doi.org/10.5061/dryad.3bk3j9kh3 | Dryad Digital Repository, 10.5061/dryad.3bk3j9kh3 |

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
