## [Editor Report]

In this study Ji and colleagues investigate the circuit mechanisms that control long lasting behavioral foraging states in the nematode *C. elegans*. In a series of elegant neuronal imaging, circuit manipulation and behavioral genetics experiments they show how populations of neurons tightly coordinate their ensemble activity to achieve a surprising degree of multi-functionality. This includes the control of locomotion parameters that characterize each foraging state, their mutual exclusiveness and persistence, as well as sensory integration for informed state switching.

---

## [Decision Letter]

**Decision letter after peer review:**

Thank you for submitting your article "A Neural Circuit for Flexible Control of Persistent Behavioral States" for consideration by *eLife*. Your article has been reviewed by 3 peer reviewers, one of whom is a member of our Board of Reviewing Editors, and the evaluation has been overseen by Piali Sengupta as the Senior Editor. The reviewers have opted to remain anonymous.

The reviewers have discussed the reviews with one another and the Reviewing Editor has drafted this decision to help you prepare a revised submission.

Summary:

An unsolved problem in neuroscience is how neuronal circuits control long lasting behavioral states. In this manuscript, Ji et al. apply multi-neuron imaging in freely behaving animals, genetics, circuit interrogation tools, machine learning and modelling to study how the nematode *C. elegans* control,s maintenance and alternations between two mutually exclusive behavioral states termed roaming and dwelling. In previous work, some of the same authors showed that 5-HT releasing NSM neurons promote dwelling, while several pigment-dispersing factor (PDF) releasing neurons promote roaming. However, how neuronal activity of these populations of neurons relates to these behavioral states was mostly unknown. Moreover, a circuit model showing how these neurons modulate their targets and thereby behavioral output was lacking. Here, the authors propose persistent population activity as a neuronal correlate of roaming and dwelling states. They propose a model of mutual inhibition between NSM and PDF receptor expressing neurons as a circuit mechanism underlying dwelling-roaming switches. Moreover, they propose an intriguing bi-functional role of AIA neurons in triggering sensory evoked roaming dwelling switches.

Essential revisions:

Please find below a summary of the reviewer's comments. While they find your work very interesting and potentially important, they raised concerns about lacking evidence for persistent circuit activity and the mutual inhibition model. Without at least one of these aspects being substantiated, the work described here appears to be represent only a small advance over previously published work, e.g. your 2013 paper Flavell et al. Therefore, we find it essential that these critiques are fully addressed in a revised manuscript.

(I) Persistent neuronal activity:*Reviewer #1:*

(1) The authors claim that persistent neuronal-circuit activity reflects roaming-dwelling changes, with particular focus on persistent NSM activity throughout dwelling. Inspecting activity traces, including also the corresponding author's previous data (Flavell, Cell, 2013), I am not convinced that this is the case. Dwelling is characterized by slow forward locomotion interspersed by frequent intermittent reversal maneuvers. Neuronal activity explicitly encoding the dwelling state, therefore should persist throughout these switches. However, NSM is rapidly fluctuating during dwelling (see Figure 1C) and, more importantly, seems to be negatively correlated with forward locomotion speed and rapidly down-regulated or inactive during reversals (Figure 1C-D). In this view, average NSM activity at dwelling onsets (Figure 1H) would reflect an average pattern in locomotion speed. A similar tight coupling of neuronal activity with instantaneous behavioral parameters is seen for most other neurons in the present study and this was shown already quantitatively in several previous studies: neuronal activity of AVB, RIB and AIY are positively correlated with forward locomotion speed and inactive during reversals; while AVA activity is tightly locked to reversals. I do not see evidence that any of these neurons' activity is persistent throughout roaming or dwelling, but they rather encode instantaneous parameters and switches in the worms ongoing locomotion. It is possible that these neurons' endocrine signaling via 5-HT and PDF effectively outlasts faster fluctuations of their Ca++ activity, but this remains speculative. I think these are crucial details and if the authors cannot convince me otherwise by more detailed analysis of their data, I think they need to revise one of the major statements they make throughout the paper. The authors would need to show that neuronal activity of NSM and other neurons persists throughout dwelling (and roaming), robustly and irrespective of speed fluctuations and forward-backward switches. Again, for AVA, AVB, RIB and AIY, previous studies that also applied Ca++ imaging in freely crawling worms showed that this is not the case.

(2) Taken into account the considerations above, PC2 seems to capture the inverse activity of NSM/AIA versus AVB, RIB, AIY, during foreword locomotion (Figure 1Suppl. 3B) with some contribution of AVA and others. In this view PC2 partially reflects the activity of neurons correlated/anti-correlated with forward locomotion speed, which overlaps with the dwelling state, but I don't see evidence for PC2 reflecting a persistent population state for dwelling. How should such a population-code get established and read-out?*Reviewer #2:*

(3) The authors suggest that stereotyped circuit-wide activity patterns corresponding to each foraging state are stereotyped but they only present representative calcium traces and ethograms (roaming vs. dwelling) for single animals. This makes it challenging to assess how stereotyped the relationships between activity and behavior are across animals, neuron types, over time, as a function of locomotor kinematics or state. It would be helpful for the authors to present the data in a simpler form (not just bar-graphs or PCA trajectories) to give the reader a better sense of the variability in this rich dataset. A useful demonstration of the stereotypical relationship between network activity and behavior, is to use the calcium activity of these 10 neurons to predict behavior or at least roaming or dwelling.

(4) The authors suggest NSM plays a unique role in organizing the circuit-wide activity state that corresponds to dwelling. To support this, it would be useful to see each of the 10 neuron's activity plotted on separate axes in PCA space, as for NSM in Figure 1G, as well as plots 1H, Figure 1-Supp3 B, and 1I across neurons for comparison.

(II) Mutual inhibition model*Reviewer #2:*

(1) AVB and NSM were barely coactive in WT, but were frequently coactive in pdfr mutants; during roaming to dwelling transition, the rise of NSM signal was accompanied by a decrease of AVB/RIB calcium signals. These results suggest that NSM is inhibited by polysynaptic inputs or neuromodulators from PDFR neurons.

I am not convinced by their argument. The lack of PDF receptor may have a broad effect on the neural circuit, and it is not clear to me why it would only affect the outputs from PDFR neurons to NSM.

In fact, an equally possible (perhaps even better) model is that the activities of NSM and PDFR neurons are differentially modulated by upstream "sensory" inputs through a push-pull mechanism, which can also elicit state transitions without involving a mutual-inhibition loop. Moreover, I postulate that, in pdfr mutants, NSM and PDFR neurons are more likely to be co-activated by sensory inputs, and the competition between the two groups of neurons prevents the animal from roaming. This alternativethe model also explains why AVB/AIY/RIB still exhibit strong and correlated activities during the dwelling state in pdfr mutants.

(2) Likewise, the argument on the inhibition from NSM to MOD-1/PDFR neurons suffers the same problem.*Reviewer #3:*

(3) From the data in Figure 2, the authors conclude that there is a mutual inhibitory relationship between mod-1 neurons and NSM. This conclusion would be better supported by explicitly plotting the relationship between NSM and MOD-1-expressing neuron activities to show an inverse correlation in WT animals, and loss of inverse-correlation in mod-1 animals. Even more direct functional evidence of this inhibitory relationship (even if polysynaptic) would require monitoring NSM activity during acute activation or inhibition of mod-1 neurons. Fundamentally, inferring causal relationships between neuromodulatory systems and circuit dynamics is challenging from examination of mutants alone, since behavioral state space is also dramatically changed in these animals. For example, tph-1 and mod-1 mutants roam more, so it is not surprising that NSM activity is decreased on average. Is there a way to normalize the neuronal activity measurements in Figure 2 to account for the altered baseline behavior?

(4) The authors suggest that NSM and AVB activities are correlated in WT conditions and become anti-correlated in pdfr-1. Cross-correlations in Figure 3-Supp1 seem to be a bit at odds with this interpretation and rather there is no apparent correlation in WT and the two neurons become anti-correlated in pdfr-1 animals.

(5) In epistasis experiments, the authors use a sort of classic logic typically applied to molecular signaling pathways to determine whether MOD-1 or PDFR-1-expressing neurons act in the same or parallel pathways. However, given the highly interconnected nature of this circuit and that both MOD-1 and PDFR-1 expressing neurons are heterogenous in their properties, activity, and connectivity patterns (only partially depicted Figure 1B), it seems challenging to use these results to ascertain whether these sub-populations are uniformly "upstream" or "downstream" of each other simply based on the experiments in Figure 3.

(6) To constrain any model of how circuit dynamics is altered during behavior, it would seem valuable for the authors to more directly probe the relationships between neurons rather than infer them from potentially pleiotropic mutants that also dramatically change behavioral state space. For example, it could be helpful to examine the relationship between NSM and MOD-1-expressing neuron activities in WT and mod-1 mutants, normalizing for the differences in behavior, as suggested (see reviewer #1) by drawing subsets of data form WT that have the same speed distribution as mutants.

Alternatively, they could monitor NSM activity during acute activation or inhibition of MOD-1 expressing neurons to more directly reveal inhibition.*Reviewer #1:*

(7) In Figure 3C the authors show how normalized (0-1) activity of NSM and AVB activities are mutually exclusive, which cannot be seen in pdfr-1 and tph-1; pdfr-1 mutants. I am concerned that this could be a result of pdfr-1 mutants being 'confined' to the subspace of low locomotion speed, and thus high NSM and low AVB activity levels. Conversely, the acy-1(gf) transgene confines the data to the high-speed sub-space. An alternative interpretation is that the relationship between NSM and AVB is stable in the various genetic backgrounds, but data are skewed by the different speed distributions. Especially in the acy-1(gf) data this seems to be the case: the distribution of orange dots seem to be identical to the same speed data in WT. The normalization steps performed by the authors could distort such stable relationships drastically. This can be tested by drawing subsets of data form WT that have the same speed distribution like pdfr-1 mutants and acy-1(gf) animals, and then redo the analyses.

(III) Other essential revisions:

(8) The authors compare their PCA results with a previous study (Kato, 2015). While PC1 in the present study recapitulates aspects of the previous work (switching between forward and backward active neurons), higher PCs receive contributions from different neurons. The authors conclude that the worm's neuronal activity space would considerably vary across environmental conditions. One cannot draw this conclusion, because different subsets of neurons were recorded in the two studies, it is thus technically wrong to pairwise compare single PCs between these very different datasets.

(9) The authors designed a CNN to predict NSM state from preceding neuronal activity. This was indeed useful as it revealed candidate regulators like AIA. But they conclude from this:

"Together, these results reveal a stereotyped, multi-neuron activity pattern that predicts NSM activation". Two neurons with strong predictive score are RIB and AVA, which have strong correlates with behavior (Speed and Reversals). Therefore, we wonder if these multi neuron activity patterns contain more information than behavioral correlates. Indeed, one expects that reversals and slow speed would predict dwelling onsets. Is the prediction power of speed and reversal significantly lower than the neuronal activity patterns?

(10) CNN classifier and machine learning.

I'm fond of the idea of predicting behavior transitions from neuronal activity data. But there are technical questions that are buried deep in the methods. For example, (1) Why use two 9-by-3 convolutional kernels? (2) Why choose 30 second time window for average? (3) If the generalized linear model works equally well as CNN (Figure 4 —figure supplement 1), why don't we follow the principle of Occam's razor?

[Editors’ note: further revisions were suggested prior to acceptance, as described below.]

Thank you for resubmitting your work entitled "A Neural Circuit for Flexible Control of Persistent Behavioral States" for further consideration by *eLife*. Your article has been reviewed by 2 peer reviewers, and the evaluation has been overseen by a Reviewing Editor and Piali Sengupta as the Senior Editor.

The reviewers discussed their comments and agreed that the manuscript has been improved but there are some remaining issues that need to be addressed. Both reviewers are not convinced that your data support one of the major conclusions of stable network states corresponding to roaming or dwelling. In this view, reviewer #2 questions the validity of the model, which should be better justified and further described in more detail.

We would be excited to publish your revised article in *eLife* if you can address the comments below.

*Reviewer #1:*

In the revised manuscript Ni Ji and colleagues made substantial efforts in addressing my comments. They show convincingly that NSM neuronal activity persists throughout dwelling episodes and that 5-HT and PDFR expressing neurons affect each other consistent with a mutual inhibition model. However, their proposal of a persistent network state directly corresponding to roaming dwelling switches is not supported by the data provided, as most neurons in this study are rather primarily locked to locomotion parameters.

(1) The new analyses indicate that NSM activity differs from other locomotion associated neurons, in a way that it relates to the overall dwelling state as opposed being just anti-correlating to forward locomotion speed. This indeed shows that some aspects of NSM activity persist throughout the dwelling state. I wonder how NSM activity alone would perform in the logistic regression model Figure 1I. If it performs worse than the neuronal population, it should be discussed why. I find the Results section around Figure 1I incomplete without this obvious additional analysis.

(2) The authors provide convincing new experimental data supporting mutual inhibition between NSM and PDFR expressing neurons, a point mostly raised by the other reviewers.

(3) I am less convinced by the authors' proposal of circuit wide network states explicitly encoding dwelling or roaming, which is not sufficiently supported by the data. It is known that activity of reverse active interneurons like AVA, when recorded in freely moving animals, gradually correlates with reverse locomotion speed (Kato et al., 2015), like the forward active interneurons (AVB, AIY, RIB) gradually correlate with forward locomotion speed (Kato et al., 2015; Li et al., 2014; Luo et al., 2014). These features extend to other interneuron classes not studied in the present manuscript (Hallinen et al., 2021; Kato et al., 2015). Hence, this is not a new finding of the present study and I feel that this vast amount of literature is not adequately cited. Crucially, since roaming-dwelling states are classified by two speed parameters (Methods), it does not come at a surprise at all that these behavioural states have a signature in network activity; again, because most of the neuronal activities under study are primarily sensitive to locomotion speed during either forward or backward crawling. This, however, should not be confused with a persistent network state that discretely switches between roaming- and dwelling correlates. There is no evidence presented that other neurons than NSM, and perhaps AIA, receive explicit additional modulations during roaming or dwelling. In this manner, the authors over-interpret a PCA result, where PC2 receives some contributions from these locomotion tuned neurons. This is exactly what you expect from PCA analysis of neurons with relationships described above. The statement in the rebuttal letter "we reveal for the first time that the activities of these neurons are also modulated by behavioral state, showing an overall decrease in activity during dwelling" is misleading in this sense. In the same vein, the authors keep their formulations somewhat ambiguous, which I think is misleading readers not familiar with the details in the literature cited above:

P5: Together, these data reveal widespread encoding of locomotion parameters and

foraging state by neurons throughout the roaming-dwelling circuit.

P22: Circuit-wide calcium imaging during roaming and dwelling identified stable activity patterns that correspond to each state….. The circuit architecture uncovered here provides new insights into how circuits generate persistent activity patterns.

P22-23: We found that NSM and AVB, which produce 5-HT and PDF-1 respectively, have mutually-exclusive activities that correlate with dwelling or roaming.

Hallinen, K.M., Dempsey, R., Scholz, M., Yu, X., Linder, A., Randi, F., Sharma, A.K., Shaevitz, J.W., and Leifer, A.M. (2021). Decoding locomotion from population neural activity in moving *C. elegans*. *eLife* 10.

Kato, S., Kaplan, H.S., Schrödel, T., Skora, S., Lindsay, T.H., Yemini, E., Lockery, S., and Zimmer, M. (2015). Global Brain Dynamics Embed the Motor Command Sequence of *Caenorhabditis elegans*. Cell 163, 1-50.

Li, Z., Liu, J., Zheng, M., and Xu, X.Z.S. (2014). Encoding of both analog- and digital-like behavioral outputs by one *C. elegans* interneuron. Cell 159, 751-765.

Luo, L., Wen, Q., Ren, J., Hendricks, M., Gershow, M., Qin, Y., Greenwood, J., Soucy, E.R., Klein, M., Smith-Parker, H.K., et al. (2014). Dynamic Encoding of Perception, Memory, and Movement in a *C. elegans* Chemotaxis Circuit. Neuron 82, 1115-1128.

*Reviewer #2:*

The authors have improved their manuscript. However, I still have major concerns on the current revision. I will stick to the questions raised in my previous review.

Mutual inhibition:

(1) The authors provide additional optogenetic experiments to support their mutual inhibition model, namely by activating specific neurons (e.g., NSM) while recording the activity in others (e.g., AVB and AIY). Given the very slow time scale of activity change (~ 10 s after laser onset in Figure 2G) and the diffusive nature of neuromodulators, I am not sure that the data directly support a causal relationship depicted in Figure 3G and Figure 4I. A more illuminating experiment is to carry out the control experiment in mod-1 mutant (the no ATR control is useful, but not instructive).

(2) Mutual exclusivity between NSM and AVB activity appears to be an oversimplification. While the joint probability – when NSM and AVB both exhibit high activity – is small (new plot in Figure 4C WT, Figure 4 —figure supplement 3A WT), this observation does not lead to the conclusion that the two neurons are mutually exclusive. Note that much of the neural activity is distributed in a low signal region, suggesting more complicated activity patterns. For example, in the dwelling state, AVB neurons could exhibit prominent calcium transients, but presumably smaller mean amplitude, see raw traces in Figure 1 —figure supplement 3. As a result, the joint probability distribution would bias towards the lower right quadrant. Overall, the writing is confusing and misleading without a clear and rigorous definition of mutual exclusivity.

(3) Mutual inhibition and the corresponding neural dynamics could occur in both slow and fast timescales. The authors mention in the Introduction that direct experimental evidence linking recurrent circuitry with persistent activity remains scarce. Here, I would like to bring to your attention two earlier papers in *C. elegans* that will complement the current study. One is Robert et.al., *eLife* 2016 (DOI: 10.7554/*eLife*.12572) and the other Wang et al., *eLife* 2020 ( DOI: 10.7554/*eLife*.56942) both studies investigate flexible motor control, and the roles of mutual inhibitions in motor state transitions.

Modelling:

As alluded in my previous review, a model would be useful if it provides additional insights that cannot be articulated by the experimental data and plain text alone. I still don't find much insight in the current revision.

(4) The idea of representing dwelling and roaming states with two group of neurons with persistent activity is not supported by their data, as reviewer #1 clearly pointed out. The model, however, was constructed based on such a simplification. They need to justify such an assumption in their writing, or extend the model with a reasonable effort. If they cannot, I would suggest that they delete the model to avoid misleading the readers.

(5) The modelling session remains poorly written. In Equations. (1)-(3), the meanings of \γ_u, \γ_a, and \γ_b are not explained. Neither does G(t) in Equation. (5). It is not clear to me whether Equations. (5)-(6) can be approximated by steady state solutions, or why they are not. And this is why timescale is important.

Whereas the data suggest that the roaming neurons' calcium activity are fluctuating much faster than NSM, the variable x and y (Equations. (5)-(6)) in their model would lead to fluctuation at the same timescale.

(6) Specifically, how and why 'AIA' would promote roaming during the odor gradient climbing is presented in a very superficial and mysterious way. It remains entirely unclear whether the "predictions" result from the general structure of the motif or fine-tuning many of the parameters in the model. I also suggest that they present a phase diagram of some of the essential model parameters that would impact the model outcome. One such parameter, for example, is \α_x and \α_y, the feedforward connection strengths.

(7) By trying to understand your model, here I tend to offer my own explanation. Given that AIA connects with both roaming and dwelling neurons in their model, the key appears to be the asymmetry in the feedforward connections, namely the synaptic weight to the roaming neuron (α_y) is greater than the weight to the dwelling neurons (α_x).

---

## [Author Response]

(I) Persistent neuronal activity:Reviewer #1:(1) The authors claim that persistent neuronal-circuit activity reflects roaming-dwelling changes, with particular focus on persistent NSM activity throughout dwelling. Inspecting activity traces, including also the corresponding author's previous data (Flavell, Cell, 2013), I am not convinced that this is the case. Dwelling is characterized by slow forward locomotion interspersed by frequent intermittent reversal maneuvers. Neuronal activity explicitly encoding the dwelling state, therefore should persist throughout these switches. However, NSM is rapidly fluctuating during dwelling (see Figure 1C) and, more importantly, seems to be negatively correlated with forward locomotion speed and rapidly down-regulated or inactive during reversals (Figure 1C-D). In this view, average NSM activity at dwelling onsets (Figure 1H) would reflect an average pattern in locomotion speed. A similar tight coupling of neuronal activity with instantaneous behavioral parameters is seen for most other neurons in the present study and this was shown already quantitatively in several previous studies: neuronal activity of AVB, RIB and AIY are positively correlated with forward locomotion speed and inactive during reversals; while AVA activity is tightly locked to reversals. I do not see evidence that any of these neurons' activity is persistent throughout roaming or dwelling, but they rather encode instantaneous parameters and switches in the worms ongoing locomotion. It is possible that these neurons' endocrine signaling via 5-HT and PDF effectively outlasts faster fluctuations of their Ca++ activity, but this remains speculative. I think these are crucial details and if the authors cannot convince me otherwise by more detailed analysis of their data, I think they need to revise one of the major statements they make throughout the paper. The authors would need to show that neuronal activity of NSM and other neurons persists throughout dwelling (and roaming), robustly and irrespective of speed fluctuations and forward-backward switches. Again, for AVA, AVB, RIB and AIY, previous studies that also applied Ca++ imaging in freely crawling worms showed that this is not the case.

To address this concern, we have characterized the dynamics of the neurons more exhaustively and have conducted further analyses to relate their activity to behavior.

For each neuron, we assessed its characteristic timescale of activation by measuring the autocorrelation function of the GCaMP signal. We observed that of all the neurons recorded, NSM activity decayed the slowest (Figure 1-Supplement 3B). This observation is consistent with NSM’s minutes-long bouts of activity whose durations are strongly correlated with the durations of co-occurring dwelling states (Figure 2C-D). We also analyzed event-triggered averages of each neuron’s activity time-locked to roam/dwell transitions and forward/reverse transitions (Figure 1D). NSM showed a robust increase in activity at dwelling onset, but did not show any change in activity time-locked to fwd/rev transitions (Figure 1D). Consistent with this observation, we also found that the overall distributions of NSM GCaMP signals are identical during rapid forward and rapid reverse locomotion, but the GCaMP signals are significantly higher during paused (slow forward/backward) movement typical of dwelling (Figure 1E and Figure 2A). (As a minor point of clarification, animals display reversals during both roaming and dwelling, but those during roaming are long reversals whereas those during dwelling are very brief and typically low velocity). These data suggest that NSM’s activity is unrelated to forward/backward locomotion changes, but is higher during slow-locomotion dwelling states and shows a time-locked, sustained increase in activity at the onset of dwelling states.

We performed similar analyses for the other neurons as well. Consistent with previous reports (see citations in the manuscript), we found that, during roaming, neurons including AVB, AIY, and RIB showed increased activity during forward runs and decreased activity during reversals, while the AVA neuron showed heightened activity specifically during reversals (Figure 1D-E and Figure 1-Supplement 5A). During dwelling, both the forward- and reverse-active neurons showed decreased activity, consistent with the suppression of rapid locomotion during dwelling (Figure 1D-E). Thus, our analyses corroborate previous studies on the roles for AIY, RIB, and AVB in promoting forward locomotion and AVA in promoting backward locomotion. Further, we reveal for the first time that the activities of these neurons are also modulated by behavioral state, showing an overall decrease in activity during dwelling.

(2) Taken into account the considerations above, PC2 seems to capture the inverse activity of NSM/AIA versus AVB, RIB, AIY, during foreword locomotion (Figure 1Suppl. 3B) with some contribution of AVA and others. In this view PC2 partially reflects the activity of neurons correlated/anti-correlated with forward locomotion speed, which overlaps with the dwelling state, but I don't see evidence for PC2 reflecting a persistent population state for dwelling. How should such a population-code get established and read-out?

As suggested, we have more closely examined the composition and dynamics of PC2 in relation with behavior.

Interestingly, PC2 receives positive loading from NSM, but negative loading from the forward- and reverse-active neurons AIY, AVA, and RIB (Figure 1–Supplement 6A). Thus, when PC2 is high, NSM tends to be active while the forward- and reverse-active neurons tend have reduced activity. In contrast, the factor loadings on PC1 are strongly positive for neurons active during forward runs (AIY, AVB, RIB, RME, etc.), small for NSM, and strongly negative for the reversal-active AVA neuron. This suggests that PC1 dynamics should correlate with animal velocity, which is confirmed in Figure 1F.

Whereas PC1 is high during forward movement and low during reverse movement (Figure 1F), PC2 is the same regardless of movement direction (fwd/rev), but is specifically active during slow movement in the dwelling state (Figure 1F-G). These results are consistent with our single neuron analyses described above in point (1) and suggest that rapid forward/backward movement is associated with neural dynamics along PC1, while the dwelling state is associated with PC2. The fact that several neurons are loaded onto both PC1 and PC2 is consistent with the fact that their activities are modulated by forward/reverse direction changes and overall behavioral state (suppressed during dwelling). We try to make these relationships clearer in the writing of our revised manuscript. (p. 7-10)

Reviewer #2:(3) The authors suggest that stereotyped circuit-wide activity patterns corresponding to each foraging state are stereotyped but they only present representative calcium traces and ethograms (roaming vs. dwelling) for single animals. This makes it challenging to assess how stereotyped the relationships between activity and behavior are across animals, neuron types, over time, as a function of locomotor kinematics or state. It would be helpful for the authors to present the data in a simpler form (not just bar-graphs or PCA trajectories) to give the reader a better sense of the variability in this rich dataset. A useful demonstration of the stereotypical relationship between network activity and behavior, is to use the calcium activity of these 10 neurons to predict behavior or at least roaming or dwelling.

We provide a more extensive set of analyses relating each neuron’s activity to behavior across animals in the revised manuscript. These plots include behavioral-event-triggered averages (Figure 1D), distributions of GCaMP signals across different behaviors (Figure 1E), and more. In addition, we examined whether neural activity in the circuit could predict (i) behavioral state and (ii) acute velocity by training two separate models (logistic regression to predict roam/dwell and a General Linear Model to predict analog velocity). We found a high level of cross-validated performance for both models (Figure 1H-I; Figure 1—figure supplement 7).

(4) The authors suggest NSM plays a unique role in organizing the circuit-wide activity state that corresponds to dwelling. To support this, it would be useful to see each of the 10 neuron's activity plotted on separate axes in PCA space, as for NSM in Figure 1G, as well as plots 1H, Figure 1-Supp3 B, and 1I across neurons for comparison.

We have provided these plots, as suggested: (i) each neuron’s activity during trajectories through PCA space (Figure 1-Supplement 6B); (ii) each neuron’s activity triggered on Fwd/Rev and Roam/Dwell transitions (Figure 1D), and (iii) each neuron’s factor loadings on PC1 and PC2 (Figure 1-Supplement 6A).

(II) Mutual inhibition modelReviewer #2:(1) AVB and NSM were barely coactive in WT, but were frequently coactive in pdfr mutants; during roaming to dwelling transition, the rise of NSM signal was accompanied by a decrease of AVB/RIB calcium signals. These results suggest that NSM is inhibited by polysynaptic inputs or neuromodulators from PDFR neurons.I am not convinced by their argument. The lack of PDF receptor may have a broad effect on the neural circuit, and it is not clear to me why it would only affect the outputs from PDFR neurons to NSM.In fact, an equally possible (perhaps even better) model is that the activities of NSM and PDFR neurons are differentially modulated by upstream "sensory" inputs through a push-pull mechanism, which can also elicit state transitions without involving a mutual-inhibition loop. Moreover, I postulate that, in pdfr mutants, NSM and PDFR neurons are more likely to be co-activated by sensory inputs, and the competition between the two groups of neurons prevents the animal from roaming. This alternativethe model also explains why AVB/AIY/RIB still exhibit strong and correlated activities during the dwelling state in pdfr mutants.(2) Likewise, the argument on the inhibition from NSM to MOD-1/PDFR neurons suffers the same problem.

To directly test whether NSM and MOD-1/PDF-1 (i.e. roaming-active) neurons mutually inhibit one another, we performed new experiments in which we optogenetically activated specific neurons while recording others. We found that NSM activation acutely inhibited the roaming-active neurons (Figure 2G). Moreover, activation of either the MOD-1-expressing neurons or the PDF-1-expressing neurons acutely inhibited NSM (Figures 3F and 4G). These experiments provide strong support for the mutual inhibitory loop that we describe, but don’t rule out the possibility of additional push-pull mechanisms elsewhere in the circuit.

Reviewer #3:(3) From the data in Figure 2, the authors conclude that there is a mutual inhibitory relationship between mod-1 neurons and NSM. This conclusion would be better supported by explicitly plotting the relationship between NSM and MOD-1-expressing neuron activities to show an inverse correlation in WT animals, and loss of inverse-correlation in mod-1 animals. Even more direct functional evidence of this inhibitory relationship (even if polysynaptic) would require monitoring NSM activity during acute activation or inhibition of mod-1 neurons. Fundamentally, inferring causal relationships between neuromodulatory systems and circuit dynamics is challenging from examination of mutants alone, since behavioral state space is also dramatically changed in these animals. For example, tph-1 and mod-1 mutants roam more, so it is not surprising that NSM activity is decreased on average. Is there a way to normalize the neuronal activity measurements in Figure 2 to account for the altered baseline behavior?

We agree that interpretation of neural activity in the mutants is potentially complex. Therefore, to directly test whether NSM and MOD-1/PDF-1 (i.e. roaming-active) neurons mutually inhibit one another in a wild-type background, we performed new experiments in which we optogenetically activated specific neurons while recording others. We found that NSM activation acutely inhibited the roaming-active neurons (Figure 2G). Moreover, activation of either the MOD-1-expressing neurons or the PDF-1expressing neurons acutely inhibited NSM (Figures 3F and 4G). These experiments provide additional experiment support for a mutual inhibitory loop.

(4) The authors suggest that NSM and AVB activities are correlated in WT conditions and become anti-correlated in pdfr-1. Cross-correlations in Figure 3-Supp1 seem to be a bit at odds with this interpretation and rather there is no apparent correlation in WT and the two neurons become anti-correlated in pdfr-1 animals.

We apologize for the ambiguity of this description in the previous draft of the manuscript. As is shown in the 2-D histogram in Figure 4C (left panel), the joint distribution of NSM and AVB activity in wild-type animals lie largely near the two axes (“L-shape”). Ideally, mutual exclusivity corresponding to a uniform, “L-shaped” distribution of data points would indeed yield a negative correlation. However, the observed NSM-AVB distribution exhibits a concentration of data points near the origin (i.e. low NSM activity and low AVB activity), when neither neuron is active. This concentration of data points near the origin, which does not exhibit any clear positive or negative correlation, significantly influences the overall correlation of the full distribution. Thus, given the sensitivity of the correlation coefficient, we instead define “mutual exclusivity” as a lack of co-activity between NSM and AVB, which is manifested by the lack of probability density at the upper right quadrant of the joint activity space (Figure 4C). In the revised manuscript, we present new visualization and quantification of the degree of co-activity across different genotypes (Figures 4C-D and Figure 4-Supplement 3).

Finally, in *pdfr-1* mutants, NSM and AVB become strikingly correlated, which we communicate better in the revised manuscript also. This can be observed using the co-activity metric described above (Figure 4D) and in the neuron-vs-neuron cross-correlation matrices that we provide in the revised manuscript (Figure 4Supplement 2).

(5) In epistasis experiments, the authors use a sort of classic logic typically applied to molecular signaling pathways to determine whether MOD-1 or PDFR-1-expressing neurons act in the same or parallel pathways. However, given the highly interconnected nature of this circuit and that both MOD-1 and PDFR-1 expressing neurons are heterogenous in their properties, activity, and connectivity patterns (only partially depicted Figure 1B), it seems challenging to use these results to ascertain whether these sub-populations are uniformly "upstream" or "downstream" of each other simply based on the experiments in Figure 3.

We agree that the epistasis experiments could be complex given the highly interconnected circuit. Therefore, to directly examine how neurons influence one another’s activity in this circuit, we performed new experiments in which we optogenetically activated specific neurons while recording others. We found that NSM activation acutely inhibited the roaming-active neurons (Figure 2G). Moreover, activation of either the MOD-1-expressing neurons or the PDF-1-expressing neurons acutely inhibited NSM (Figures 3F and 4G). These experiments provide experiment support for a mutual inhibitory loop. Regarding the epistasis experiments specifically, we have softened the language in our conclusions from these experiments and pointed out the complexity highlighted by the reviewer (p. 17).

(6) To constrain any model of how circuit dynamics is altered during behavior, it would seem valuable for the authors to more directly probe the relationships between neurons rather than infer them from potentially pleiotropic mutants that also dramatically change behavioral state space. For example, it could be helpful to examine the relationship between NSM and MOD-1-expressing neuron activities in WT and mod-1 mutants, normalizing for the differences in behavior, as suggested (see reviewer #1) by drawing subsets of data form WT that have the same speed distribution as mutants.Alternatively, they could monitor NSM activity during acute activation or inhibition of MOD-1 expressing neurons to more directly reveal inhibition.

As is described above, we have performed a series of optogenetics experiments to directly validate the mutual inhibitory interactions between NSM and the MOD-1- and PDF-expressing neurons (Figures 2G, 3F, and 4G).

Reviewer #1:(7) In Figure 3C the authors show how normalized (0-1) activity of NSM and AVB activities are mutually exclusive, which cannot be seen in pdfr-1 and tph-1; pdfr-1 mutants. I am concerned that this could be a result of pdfr-1 mutants being 'confined' to the subspace of low locomotion speed, and thus high NSM and low AVB activity levels. Conversely, the acy-1(gf) transgene confines the data to the high-speed sub-space. An alternative interpretation is that the relationship between NSM and AVB is stable in the various genetic backgrounds, but data are skewed by the different speed distributions. Especially in the acy-1(gf) data this seems to be the case: the distribution of orange dots seem to be identical to the same speed data in WT. The normalization steps performed by the authors could distort such stable relationships drastically. This can be tested by drawing subsets of data form WT that have the same speed distribution like pdfr-1 mutants and acy-1(gf) animals, and then redo the analyses.

We performed further analyses to examine this concern. First, we examined the range and distribution of GCaMP fluorescence measurements from NSM and AVB in the mutants that we recorded (using F/Fmean values, thus avoiding the (0-1) normalization method that raised the concerns). We observed that there was no difference across the mutants, indicating that the range of GCaMP signals observed in each experimental condition is similar (Figure 4-Supplement 3D-E). There was one notable exception to this: the NSM GCaMP distribution was narrowed in the pdfr-1::acy-1(gf) mutants – this indicates that NSM GCaMP dynamics are reduced in this genetic background.

We also performed the analysis suggested by the reviewer where we compared NSM/AVB co-activity across mutants, drawing subsets of data that are matched for speed distributions. This analysis confirmed that the mutual exclusivity between NSM and AVB can be detected when sampling from data points with similar speed distributions and that this mutual exclusivity is disrupted in *pdfr-1* mutants (Figure 4Supplement 3A-B).

Finally, because we agree that the (0-1) normalization could in principle lead to the types of effects that the reviewer raised, we provide new figure panels showing that the key NSM/AVB results are essentially the same when using F/Fmean instead (Figure 4-Supplement 3A-B).

(III) Other essential revisions:(8) The authors compare their PCA results with a previous study (Kato, 2015). While PC1 in the present study recapitulates aspects of the previous work (switching between forward and backward active neurons), higher PCs receive contributions from different neurons. The authors conclude that the worm's neuronal activity space would considerably vary across environmental conditions. One cannot draw this conclusion, because different subsets of neurons were recorded in the two studies, it is thus technically wrong to pairwise compare single PCs between these very different datasets.

We have adjusted the writing to remove this conclusion, as suggested by the reviewer.

(9) The authors designed a CNN to predict NSM state from preceding neuronal activity. This was indeed useful as it revealed candidate regulators like AIA. But they conclude from this:“Together, these results reveal a stereotyped, multi-neuron activity pattern that predicts NSM activation”. Two neurons with strong predictive score are RIB and AVA, which have strong correlates with behavior (Speed and Reversals). Therefore, we wonder if these multi neuron activity patterns contain more information than behavioral correlates. Indeed, one expects that reversals and slow speed would predict dwelling onsets. Is the prediction power of speed and reversal significantly lower than the neuronal activity patterns?

We attempted to train a classifier to predict NSM activation from preceding behavioral data, but were unable to obtain any classifier with performance greater than that expected by chance. Given that this negative result is somewhat inconclusive, we have decided not to include this finding in the revised manuscript.

(10) CNN classifier and machine learning.I’m fond of the idea of predicting behavior transitions from neuronal activity data. But there are technical questions that are buried deep in the methods. For example, (1) Why use two 9-by-3 convolutional kernels? (2) Why choose 30 second time window for average? (3) If the generalized linear model works equally well as CNN (Figure 4 —figure supplement 1), why don’t we follow the principle of Occam’s razor?

We have provided clearer rationale for these technical questions in the revised manuscript. Briefly, (1) the CNN architecture could in principle recover multiple convolutional kernels that each provide important information for predicting NSM activation. Thus, we used 2 convolutional kernels to allow for this possibility. In practice, we only recovered one informative kernel (the other was almost entirely comprised of values close to zero). (2) We systematically varied the width of this time window and found that the 30s averaging time window led to marginally higher performance than other time windows (results for a range of time windows are now provided in Figure 5-Supplement 1). (3) We agree that the GLM and CNN perform similarly, but these models are equally interpretable (and, thus, equally useful to make inferences about how neural activity predicts behavior), so we decided to present the analysis approach that was more innovative (and ultimately more flexible for more complex applications) in the hopes that it might be of interest to readers. These additional details about the Methods are now on p. 49-50.

[Editors’ note: further revisions were suggested prior to acceptance, as described below.]

Reviewer #1:In the revised manuscript Ni Ji and colleagues made substantial efforts in addressing my comments. They show convincingly that NSM neuronal activity persists throughout dwelling episodes and that 5-HT and PDFR expressing neurons affect each other consistent with a mutual inhibition model. However, their proposal of a persistent network state directly corresponding to roaming dwelling switches is not supported by the data provided, as most neurons in this study are rather primarily locked to locomotion parameters.(1) The new analyses indicate that NSM activity differs from other locomotion associated neurons, in a way that it relates to the overall dwelling state as opposed being just anti-correlating to forward locomotion speed. This indeed shows that some aspects of NSM activity persist throughout the dwelling state. I wonder how NSM activity alone would perform in the logistic regression model Figure 1I. If it performs worse than the neuronal population, it should be discussed why. I find the Results section around Figure 1I incomplete without this obvious additional analysis.

We performed this analysis suggested by the reviewer (Figure 1I). Briefly, we found that such a classifier indeed has very high performance, but is not quite as good as the full neural population. This is somewhat expected because we previously observed that NSM activity very reliably predicts dwelling onset, but the decline in NSM activity is not always perfectly time locked to the end of dwelling states (Figure 2C, discussed in the Discussion of manuscript). This would then be predicted to lead to a small number of errors in classifier performance.

(2) The authors provide convincing new experimental data supporting mutual inhibition between NSM and PDFR expressing neurons, a point mostly raised by the other reviewers.

Thank you.

(3) I am less convinced by the authors’ proposal of circuit wide network states explicitly encoding dwelling or roaming, which is not sufficiently supported by the data. It is known that activity of reverse active interneurons like AVA, when recorded in freely moving animals, gradually correlates with reverse locomotion speed (Kato et al., 2015), like the forward active interneurons (AVB, AIY, RIB) gradually correlate with forward locomotion speed (Kato et al., 2015; Li et al., 2014; Luo et al., 2014). These features extend to other interneuron classes not studied in the present manuscript (Hallinen et al., 2021; Kato et al., 2015). Hence, this is not a new finding of the present study and I feel that this vast amount of literature is not adequately cited. Crucially, since roaming-dwelling states are classified by two speed parameters (Methods), it does not come at a surprise at all that these behavioural states have a signature in network activity; again, because most of the neuronal activities under study are primarily sensitive to locomotion speed during either forward or backward crawling. This, however, should not be confused with a persistent network state that discretely switches between roaming- and dwelling correlates. There is no evidence presented that other neurons than NSM, and perhaps AIA, receive explicit additional modulations during roaming or dwelling. In this manner, the authors over-interpret a PCA result, where PC2 receives some contributions from these locomotion tuned neurons. This is exactly what you expect from PCA analysis of neurons with relationships described above. The statement in the rebuttal letter “we reveal for the first time that the activities of these neurons are also modulated by behavioral state, showing an overall decrease in activity during dwelling” is misleading in this sense. In the same vein, the authors keep their formulations somewhat ambiguous, which I think is misleading readers not familiar with the details in the literature cited above:

We thank the reviewer for bringing up this important point and apologize for our lack of clarity on this issue. Regarding the forward- and reverse-active neurons, we agree with the reviewer that the observation that several of these neurons display diminished activity during dwelling does not change our understanding of how these forward- and reverse-active neurons control behavior: these results are consistent with the known roles of these neurons in controlling forward and reverse movement and our manuscript does not modify this understanding. We certainly did not mean to imply this and have modified the language throughout the manuscript to ensure that this is clear. In addition, we added the suggested citations. (p. 6-7, p. 22, and elsewhere) The result that we show in the paper is that several of the forward- and reverse-active neurons show reduced activity during dwelling (overall and/or surrounding moments of state transitions; Figure 1D-E). We have included new 2-D histograms of GCaMP signals of RIB and AVA neurons (as exemplar forward/reverse neurons) during roaming vs dwelling to make these results clearer in the revised manuscript (Figure 1 – Figure Supplement 6). We believe it is important to communicate these results, as they provide insights into how neural circuit activity changes across these states. We agree that diminished activity of locomotion-related neurons during the state with less locomotion is not particularly surprising, but we note that this was not the only possible outcome. For example, before we had collected these data, we had considered an alternative hypothesis that during dwelling states the forward- and reverse-active neurons might remain highly active, but have altered coupling to behavior such that they were unable to drive robust forward and backward movement during dwelling (perhaps due to a change in downstream circuits, for example in the ventral cord). However, given our findings, we have now modified the text to make clear that this aspect of our results is not especially surprising. (p. 7 and elsewhere)

Finally, we have modified the text to limit our discussion of “persistent activity changes” to the NSM neuron, which has more prolonged bouts of activity. As examples, we indicate here how we modified the sentences that concerned the reviewer (though see the revised manuscript for similar changes in other parts of the text; under Track Changes):

P5: Together, these data reveal widespread encoding of locomotion parameters and foraging state by neurons throughout the roaming-dwelling circuit

Changed to: “Together, these data reveal that changes in the roaming/dwelling state of the animal are accompanied by changes in the activities of multiple neurons, including NSM and a set of neurons that have previously been shown to control forward and reverse locomotion.” (p. 7)

P22: Circuit-wide calcium imaging during roaming and dwelling identified stable activity patterns that correspond to each state…..

Changed to: “Circuit-wide calcium imaging during roaming and dwelling identified several neurons whose activities differ between the two states, most notably the serotonergic NSM neuron that displays long bouts of activity during dwelling and inactivity during roaming.” (p.21)

The circuit architecture uncovered here provides new insights into how circuits generatepersistent activity patterns

Changed to: “The recordings in this study provide new insights into how neural activity changes as animals switch between stable, alternative behavioral states.” (p. 21)

P22-23: We found that NSM and AVB, which produce 5-HT and PDF-1 respectively, have mutually-exclusive activities that correlate with dwelling or roaming.

Changed to: “We found that NSM displayed long bouts of activity during dwelling and was inactive during roaming. In addition, several neurons that were previously shown to drive forward and reverse movement, including the PDF-producing neuron AVB, were more active during roaming than they were during dwelling. However, whereas NSM displayed long bouts of persistent activity during dwelling, the locomotion neurons displayed fast timescale dynamics associated with forward and reverse movements during roaming, matching their previously described roles in controlling locomotion.” (p. 21-22)

Reviewer #2:The authors have improved their manuscript. However, I still have major concerns on the current revision. I will stick to the questions raised in my previous review.Mutual inhibition:(1) The authors provide additional optogenetic experiments to support their mutual inhibition model, namely by activating specific neurons (e.g., NSM) while recording the activity in others (e.g., AVB and AIY). Given the very slow time scale of activity change (~ 10 s after laser onset in Figure 2G) and the diffusive nature of neuromodulators, I am not sure that the data directly support a causal relationship depicted in Figure 3G and Figure 4I. A more illuminating experiment is to carry out the control experiment in mod-1 mutant (the no ATR control is useful, but not instructive).

We have run this suggested experiment, which can be found in Figure 2 —figure supplement 1C. Briefly, we observed that the effect of NSM::Chrimson stimulation on circuit activity was strongly attenuated in a mod-1 mutant background.

(2) Mutual exclusivity between NSM and AVB activity appears to be an oversimplification. While the joint probability – when NSM and AVB both exhibit high activity – is small (new plot in Figure 4C WT, Figure 4 —figure supplement 3A WT), this observation does not lead to the conclusion that the two neurons are mutually exclusive. Note that much of the neural activity is distributed in a low signal region, suggesting more complicated activity patterns. For example, in the dwelling state, AVB neurons could exhibit prominent calcium transients, but presumably smaller mean amplitude, see raw traces in Figure 1 —figure supplement 3. As a result, the joint probability distribution would bias towards the lower right quadrant. Overall, the writing is confusing and misleading without a clear and rigorous definition of mutual exclusivity

We thank the reviewer for bringing up this important point and apologize for our lack of clarity on this issue. We have addressed this concern in two ways: (1) First, when we use the term “mutual exclusivity” in the Results section for the first time we now define it quantitatively and point the reader to figure panels and a Methods section that explains and supports this quantification method (p. 12). Briefly, we used the Otsu method to determine the thresholds between low/high activity for AVB and NSM and now provide a description of this in the Methods of the revised manuscript (p. 39). (2) We agree that this quantification that examines the joint high activities of NSM and AVB (i.e. percent data in upper right quadrant) still leaves other aspects of NSM-AVB co-activity unexplored. As the reviewer points out, perhaps lower amplitude AVB activity during dwelling would be interesting to compare across conditions. Therefore, we also adopted a more continuous metric for measuring AVB activity, conditional on NSM activity, in which we quantified the probability of observing AVB calcium transients beyond a range of different threshold amplitudes when NSM activity is high (Figure 4 —figure supplement 2). This analysis shows the same main result, but does so in a more continuous manner: animals lacking *pdfr-1* have higher AVB activity when NSM is high. We believe that providing all of these quantification methods together now provides a more complete view of NSM-AVB co-activity.

(3) Mutual inhibition and the corresponding neural dynamics could occur in both slow and fast timescales. The authors mention in the Introduction that direct experimental evidence linking recurrent circuitry with persistent activity remains scarce. Here, I would like to bring to your attention two earlier papers in *C. elegans* that will complement the current study. One is Robert et.al., eLife 2016 (DOI: 10.7554/eLife.12572) and the other Wang et al., eLife 2020 ( DOI: 10.7554/eLife.56942) both studies investigate flexible motor control, and the roles of mutual inhibitions in motor state transitions.

We have changed this statement in our introduction to indicate that there is evidence to support a role for recurrent circuitry on this faster timescale, citing Robert et al., and Wang et al. (p. 3). In addition, we now discuss this interesting comparison of fast timescale behavioral switching and slow timescale switching in the revised Discussion (citing Robert et al; Wang et al; also Kaplan et al., *Neuron*, 2020; p. 25)

Modelling:As alluded in my previous review, a model would be useful if it provides additional insights that cannot be articulated by the experimental data and plain text alone. I still don't find much insight in the current revision.(4) The idea of representing dwelling and roaming states with two group of neurons with persistent activity is not supported by their data, as reviewer #1 clearly pointed out. The model, however, was constructed based on such a simplification. They need to justify such an assumption in their writing, or extend the model with a reasonable effort. If they cannot, I would suggest that they delete the model to avoid misleading the readers.

We had included the model in this highly simplified form because we thought that it might help readers build intuition about how AIA’s dual outputs are able to modulate behavior. However, we agree that this oversimplification results in the model not being as strongly tied to the biological circuit. Based on the reviewer feedback that this model doesn’t provide much insight, we have taken their suggestion to remove it from the paper. Its only purpose was to make things clearer, but it does seem to be sufficiently confusing (and different from the biological system) that it ends up distracting rather than helping. We also note that the model was an extremely minor component of our study, not even mentioned in the abstract or discussed in the Discussion.

(5) The modelling session remains poorly written. In Equations. (1)-(3), the meanings of \γ_u, \γ_a, and \γ_b are not explained. Neither does G(t) in Equation. (5). It is not clear to me whether Equations. (5)-(6) can be approximated by steady state solutions, or why they are not. And this is why timescale is important.

As is described above, we have removed the model from the paper, so points (5) – (7) are no longer relevant to address.